# Finding Most Influential Sets

**Lucas D. Konrad** [* 1]   **Nikolas Kuschnig** [* 2]

## Abstract

Identifying *most influential sets* (MIS) — size-$k$ subsets whose removal maximally changes a target estimand — is typically infeasible because it requires searching over $\binom{n}{k}$ subsets. For estimands with linear-fractional leave-set-out effects, we show that MIS selection reduces to a one-parameter sequence of top-$k$ problems. Dinkelbach's method yields an algorithm with $\mathcal{O}(n)$ cost per iteration and finite termination. For fixed residualized inputs, the algorithm returns a globally optimal set for the univariate ratio objective, including the oracle-residualized partial linear model. With estimated nuisance functions, uniform denominator and generated-score stability imply approximation to the first-order oracle orthogonal-score objective; exact set recovery follows under a separation condition. Simulations and applications show that the method recovers exact MIS that were previously computationally inaccessible.

## 1. Introduction

Most influential sets (MIS) are subsets of the training data whose removal induces the largest change in a target quantity of interest, such as a regression coefficient, treatment effect, or prediction. Formally, for a target functional $\phi$ and a leave-set-out estimator, the size-$k$ MIS, $\mathbb{S}_k^{\max}$, maximizes the leave-set-out discrepancy over all subsets $\mathbb{S}$ with $|\mathbb{S}| = k$. Set influence differs qualitatively from singleton influence because observations interact; some points reinforce each other, while others mask each others' effects. MIS therefore help diagnose which subsamples drive inferences and how models amplify or suppress particular groups.

In applied work, MIS provide target-specific diagnostics for interpretability, accountability, fairness, robustness, and data curation (Barocas et al., 2023; Chhabra et al., 2023; Ghorbani & Zou, 2019; Rudin, 2019; Hudgens & Halloran, 2008; Souly et al., 2025; Côté et al., 2024; Mirzasoleiman et al., 2020).

Despite their relevance, the study of MIS has historically been constrained by computation. Exact identification requires maximizing over $\binom{n}{k}$ subsets, which is computationally infeasible, even for moderate datasets. Classical influence diagnostics focus on individual data points (Cook, 1979), and practical extensions to sets are largely insensitive to higher-order interactions (Chatterjee & Hadi, 1986). Consequently, the literature has often favored robust estimation — controlling rather than identifying worst-case sensitivity (Huber & Ronchetti, 2009).

Recent work on influence functions — infinitesimal approximations that are widely used as diagnostic tools (Hampel, 1974; Koh & Liang, 2017) — has renewed interest in MIS. This has yielded influence-function-based approximations (Broderick et al., 2023), greedy heuristics and failure modes (Hu et al., 2024; Huang et al., 2025; Kuschnig et al., 2021), influence bounds (Moitra & Rohatgi, 2023; Freund & Hopkins, 2023; Rubinstein & Hopkins, 2025), and formal tests for excessive influence (Konrad & Kuschnig, 2026). However, influence functions remain inaccurate for extreme influence and for sets of data points (Basu et al., 2020; Koh et al., 2019), and efficient and accurate methods for MIS selection have remained unavailable.

In this paper, we develop an efficient algorithm for size-$k$ MIS selection for influence targets whose leave-set-out effects admit a ratio representation. The key is that these effects can be written as linear-fractional functions of the removed set, transforming the combinatorial search into a one-dimensional parametric optimization problem. Using Dinkelbach (1967)'s method, each subproblem reduces to selecting the top-$k$ scores, giving $\mathcal{O}(n)$ cost per iteration and finite termination. For fixed residualized inputs, the algorithm returns a globally optimal set for the corresponding univariate ratio objective. In partial linear models, this gives exact oracle MIS selection when the nuisance functions are known. With estimated nuisance functions, Neyman orthogonality supports first-stage stability; a separation condition yields consistent selection of the oracle MIS.

---
[*]Equal contribution  [1]Vienna University of Economics and Business, Austria [2]Monash University, Australia. Correspondence to: Nikolas Kuschnig <nikolas.kuschnig@monash.edu>.

*Proceedings of the 43rd International Conference on Machine Learning*, Seoul, South Korea. PMLR 306, 2026. Copyright 2026 by the author(s).

## 1.1. Contributions

We make four contributions.

- **Reduction.** We identify a class of leave-set-out influence objectives with linear-fractional structure and show that size-$k$ MIS selection for these reduces to a one-parameter sequence of top-$k$ selection problems.

- **Algorithm.** We apply Dinkelbach's method to obtain a finite-step algorithm with $\mathcal{O}(n)$ cost per iteration. For fixed residualized inputs, the algorithm returns a globally optimal set for the univariate ratio objective.

- **Theory.** The algorithm is exact for any fixed residualized inputs. In partial linear models, the scaled objective uniformly approximates a first-order oracle orthogonal-score objective under generated-score and denominator stability; value consistency and exact set recovery follow under a separation condition.

- **Evidence.** We benchmark the method against enumeration and greedy baselines and apply it to randomized experiments and datasets from machine learning, biology, economics, and statistics, recovering MIS at scales where enumeration is infeasible.

## 1.2. Outline

The remainder of this paper is organized as follows. Section 2 introduces the partial linear setting and formalizes the MIS problem. Section 3 develops the linear-fractional reduction, presents the algorithm, and states the theoretical guarantees. Section 4 evaluates the method in simulations and applications. Section 5 discusses relevance, limitations, and implications, and Section 6 concludes.[1]

# 2. Partial Linear Setting

We study influential sets using residualization in partial linear models (PLM; see Chernozhukov et al., 2018; Robinson, 1988). This section introduces the model and estimator, then formalizes influence and $k$-most influential sets.

## 2.1. Model and Estimation

Consider the PLM

$$y_i = x_i \beta_0 + g_0(Z_i) + u_i, \quad \mathbb{E}[u_i \,|\, x_i, Z_i] = 0, \quad (1)$$

where $(y_i, x_i, Z_i)_{i=1}^n$ are i.i.d. draws from $P$, $x_i \in \mathbb{R}$ is the treatment, $\beta_0 \in \mathbb{R}$ is the parameter of interest, and $g_0 : \mathbb{R}^d \to \mathbb{R}$ is an unknown nuisance function of covariates $Z_i \in \mathbb{R}^d$. Assume $\mathbb{E}[u_i^2 \,|\, Z_i] = \sigma^2(Z_i) < \infty$.

---

[1]The Appendix contains proofs and additional details; code is provided at https://github.com/nk027/findingMIS.

Let $h_0(Z_i) \coloneqq \mathbb{E}[x_i \mid Z_i]$, write $x_i = h_0(Z_i) + v_i$, and define $m_0(Z_i) \coloneqq \mathbb{E}[y_i \,|\, Z_i] = g_0(Z_i) + \beta_0 h_0(Z_i)$.

We estimate $\beta_0$ using residualized outcomes and treatments:

$$\tilde{y}_i \coloneqq y_i - \hat{m}(Z_i), \qquad \tilde{x}_i \coloneqq x_i - \hat{h}(Z_i), \qquad (2)$$

where $(\hat{m}, \hat{h})$ are flexible cross-fitting first-stage estimators. The residualized (Robinson) second-stage estimator is the ordinary least squares (OLS) coefficient from regressing $\tilde{y}_i$ on $\tilde{x}_i$,

$$\hat{\beta} \coloneqq \arg\min_{\beta \in \mathbb{R}} \sum_{i=1}^n (\tilde{y}_i - \tilde{x}_i \beta)^2 = \frac{\sum_{i=1}^n \tilde{x}_i \tilde{y}_i}{\sum_{i=1}^n \tilde{x}_i^2}. \qquad (3)$$

This estimator exists when $\sum_{i=1}^n \tilde{x}_i^2 > 0$; otherwise a ridge-stabilized version can be used.

## 2.2. Influence and Most Influential Sets

Let $[n] \coloneqq \{1, \ldots, n\}$. For any index set $\mathbb{S} \subseteq [n]$, let $\hat{\beta}_{-\mathbb{S}}$ denote the residualized estimator after dropping $\mathbb{S}$ from the second stage and recomputing on $[n] \setminus \mathbb{S}$.

**Definition** (Influence). *The influence of a set $\mathbb{S}$ on a scalar target $\phi : \mathbb{R} \to \mathbb{R}$ is*

$$\Delta(\mathbb{S}; \phi) = \phi\left(\hat{\beta}\right) - \phi\left(\hat{\beta}_{-\mathbb{S}}\right).$$

The target functional determines the direction and scale of change. We focus on signed affine targets, $\phi(\beta) = a\beta + b$. Because $b$ cancels and non-zero $a$ only rescales, and possibly reverses, the ordering of sets, we set $a = 1$ in the exposition. Targets with $a < 0$ correspond to directed decreases, while predicted-value targets at treatment level $x_0$ correspond to $a = x_0$.

**Definition** (Most Influential Set). *For $k < n$, a size-$k$ most influential set is any maximizer*

$$\mathbb{S}_k^{\max} \in \arg\max_{\mathbb{S} \subset [n], |\mathbb{S}| = k} \Delta(\mathbb{S}; \phi),$$

*with maximum influence $\Delta_k^{\max} \coloneqq \Delta(\mathbb{S}_k^{\max}; \phi).$*

The influence of $\mathbb{S}$ on $\hat{\beta}$ in Equation (3) admits a ratio form.

**Proposition 1** (Leave-set-out Ratio Form). *Fixing $(\tilde{x}_i, \tilde{y}_i)_{i=1}^n$, the influence of $\mathbb{S}$ on $\hat{\beta}$ is*

$$\hat{\beta} - \hat{\beta}_{-\mathbb{S}} = \frac{\sum_{s \in \mathbb{S}} \tilde{x}_s \tilde{r}_s}{\sum_{t \notin \mathbb{S}} \tilde{x}_t^2},$$

*where $\tilde{r}_s \coloneqq \tilde{y}_s - \hat{\beta}\tilde{x}_s.$*

*Proof.* The full-sample normal equation gives $\sum_{i=1}^n \tilde{x}_i \tilde{y}_i = \hat{\beta} \sum_{i=1}^n \tilde{x}_i^2$. Subtracting the contribution of

observations in $\mathbb{S}$ yields

$$\sum_{i\notin\mathbb{S}} \tilde{x}_i\tilde{y}_i = \hat{\beta}\sum_{i\notin\mathbb{S}} \tilde{x}_i^2 + \sum_{s\in\mathbb{S}} \tilde{x}_s(\hat{\beta}\tilde{x}_s - \tilde{y}_s).$$

Using $\tilde{r}_s = \tilde{y}_s - \hat{\beta}\tilde{x}_s$,

$$\sum_{i\notin\mathbb{S}} \tilde{x}_i\tilde{y}_i = \hat{\beta}\sum_{i\notin\mathbb{S}} \tilde{x}_i^2 - \sum_{s\in\mathbb{S}} \tilde{x}_s\tilde{r}_s.$$

Dividing by $\sum_{i\notin\mathbb{S}} \tilde{x}_i^2$ gives

$$\hat{\beta}_{-\mathbb{S}} = \hat{\beta} - \frac{\sum_{s\in\mathbb{S}} \tilde{x}_s\tilde{r}_s}{\sum_{i\notin\mathbb{S}} \tilde{x}_i^2},$$

which proves the claim (see Konrad & Kuschnig, 2026, for the multivariate generalization). □

This is an exact finite-sample deletion identity, and not an infinitesimal influence-function approximation. It separates two mechanisms; the numerator

$$W(\mathbb{S}) \coloneqq \sum_{s\in\mathbb{S}} \tilde{x}_s\tilde{r}_s$$

is the score removed from the full-sample normal equation, while the denominator

$$G(\mathbb{S}) \coloneqq \sum_{i\notin\mathbb{S}} \tilde{x}_i^2$$

is the residual curvature that remains after removal. Thus, a set can be influential either because it removes observations with large aligned scores, or because it removes high-curvature observations that amplify the effect of the remaining score imbalance.

Although $W(\mathbb{S})$ and $G(\mathbb{S})$ are additive separately, their ratio is not. For $i \notin \mathbb{S}$,

$$\frac{W(\mathbb{S}) + w_i}{G(\mathbb{S}) - c_i} - \frac{W(\mathbb{S})}{G(\mathbb{S})} = \frac{w_iG(\mathbb{S}) + c_iW(\mathbb{S})}{G(\mathbb{S})\{G(\mathbb{S}) - c_i\}},$$

where $w_i = \tilde{x}_i\tilde{r}_i$ and $c_i = \tilde{x}_i^2$. The first term reflects $i$'s direct score contribution; the second depends on the score already accumulated by $\mathbb{S}$. These interactions generate two phenomena. First, influence can be *joint*: observations with limited singleton influence may become highly influential when removed together. Second, influence can be *masked*: observations may appear unimportant in singleton rankings because their effect emerges only after other observations are removed. We use these terms informally in the main text; formal definitions of joint influence, masking, and greedy failure are given in Section A3.

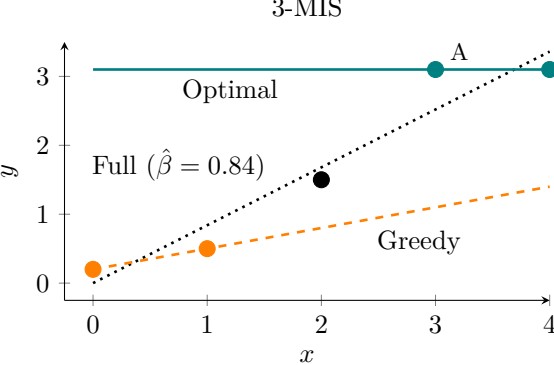

Figure 1. Greedy failure when finding the 3-MIS for $n = 5$. Point A is the most influential singleton and is therefore selected first by a greedy procedure. The exact 3-MIS instead consists of the three leftmost points, whose influence is joint and masked in singleton rankings. Enumeration recovers the optimal leave-set-out slope of 0 (solid teal line), while greedy selection is trapped on a suboptimal path.

## 2.3. Limitations of Existing Approaches

Standard methods for identifying MIS face obstacles:[2]

- *Enumeration* is computationally prohibitive. Exact search requires evaluating $\binom{n}{k}$ sets; for example, evaluating $\binom{100}{10}$ sets would take roughly 200 days at $1\mu s$ per set, and $k = 11$ would increase this to roughly $4.5$ years.

- *First-order approximations*, including influence functions (Broderick et al., 2023), are efficient but rank observations using local or singleton information, and can therefore miss joint influence and masking effects.

- *Greedy selection* (Kuschnig et al., 2021) builds a set sequentially by adding the observation with the largest current marginal gain. This implicitly assumes nested influential sets across $k$. Here, however, marginal gains depend on the current set through both $W(\mathbb{S})$ and $G(\mathbb{S})$, and early choices can exclude the optimal size-$k$ set.

**Example** (Greedy Failure Mode). *Figure 1 illustrates the resulting failure mode. Point A is the most influential singleton, so a greedy algorithm selects it first. After this choice, the search is restricted to sets containing A. The optimal 3-MIS, however, consists of the three leftmost points. These points are masked in singleton rankings — none is as influential as A on its own. Their influence is joint, emerging only when they are removed together; collectively, they accumulate enough score and remove enough curvature to flatten the fitted line. The greedy path cannot remove A after selecting it, and cannot recover the optimal size-3 set.*

---

[2]See the discussion in Kuschnig et al. (2021); Hu et al. (2024); Huang et al. (2025) for further discussion.

# 3. Method

Finding MIS requires balancing additive score contributions in the numerator against amplification from the shrinking denominator in Proposition 1. Conditional on fixed residualized inputs, this trade-off yields a linear-fractional optimization problem over subsets.

## 3.1. Linear-fractional Reduction

Map the influence components in Proposition 1 to weights and costs,

$$w_i := \tilde{x}_i \tilde{r}_i, \qquad c_i := \tilde{x}_i^2, \qquad T := \sum_{i=1}^n c_i.$$

For any set $\mathbb{S}$ of size $k$,

$$\hat{\beta} - \hat{\beta}_{-\mathbb{S}} = \frac{W(\mathbb{S})}{G(\mathbb{S})},$$

where $W(\mathbb{S}) := \sum_{i \in \mathbb{S}} w_i$, and $G(\mathbb{S}) := T - \sum_{i \in \mathbb{S}} c_i$.

**Assumption 1** (Positive denominator). *For the subset size $k$ under consideration,*

$$G(\mathbb{S}) > 0 \qquad \text{for all } \mathbb{S} \subset [n] \text{ with } |\mathbb{S}| = k.$$

Assumption 1 ensures that the estimator and fractional objective are well defined on the feasible class. If the condition fails for some sets, the same reduction applies to a ridge-stabilized denominator, replacing $G(\mathbb{S})$ by $G(\mathbb{S}) + \lambda$ for $\lambda > 0$.

## 3.2. Algorithm and Complexity

To maximize the fixed-input ratio over size-$k$ sets, we use Dinkelbach (1967)'s method. For an auxiliary parameter $\eta$, define

$$F_\eta(\mathbb{S}) := W(\mathbb{S}) - \eta\, G(\mathbb{S}) \tag{4}$$

For fixed $\eta$,

$$F_\eta(\mathbb{S}) = \sum_{i \in \mathbb{S}} (w_i + \eta c_i) - \eta T,$$

so the term $-\eta T$ is constant in $\mathbb{S}$. Maximizing $F_\eta(\mathbb{S})$ over $|\mathbb{S}| = k$ is therefore equivalent to selecting the $k$ largest scores

$$s_i(\eta) := w_i + \eta c_i.$$

Algorithm 1 alternates between this top-$k$ selection and the ratio update

$$\eta^{(t+1)} = \frac{W(S^{(t+1)})}{G(S^{(t+1)})}.$$

With exact arithmetic, finite termination occurs when the selected set no longer changes, equivalently when the Dinkelbach residual is zero. The tolerance $\tau$ is used as a numerical stopping rule.

---

**Algorithm 1** Fractional $k$-MIS selection

**Require:** weights $(w_i, c_i)_{i=1}^n$, size $k$, tolerance $\tau$,
**Require:** (optional) seed $\eta^{(0)} \leftarrow 0, S^0 \leftarrow \varnothing$
 1: Cache: $T \leftarrow \sum_{i=1}^n c_i$
 2: **for** $t = 0, 1, 2, \ldots$ **do**
 3:     Scores: $s_i \leftarrow w_i + \eta^{(t)} c_i$ for all $i$
 4:     Set: $S^{(t+1)} \leftarrow \text{TopK}(s, k)$
 5:     Numerator: $W^{(t+1)} \leftarrow \sum_{i \in S^{(t+1)}} w_i$
 6:     Denominator: $G^{(t+1)} \leftarrow T - \sum_{i \in S^{(t+1)}} c_i$
 7:     Update: $\eta^{(t+1)} \leftarrow W^{(t+1)}/G^{(t+1)}$
 8:     **if** $S^{(t+1)} = S^{(t)}$ **or** $|\eta^{(t+1)} - \eta^{(t)}| < \tau$ **then**
 9:         **return** $S^{(t+1)}, \eta^{(t+1)}$
10:     **end if**
11: **end for**

---

**Warm-start** At the optimum, the Dinkelbach parameter equals the maximal fixed-input influence,

$$\eta^\star = \max_{|\mathbb{S}|=k} \frac{W(\mathbb{S})}{G(\mathbb{S})}.$$

This identity can be used to reduce iterations when computing MIS paths across subset sizes. When tracing $k = 1, \ldots, K$, Algorithm 2 initializes the run for size $k$ at the solution value from size $k - 1$.

---

**Algorithm 2** Fractional MIS selection for $1, \ldots, K$

**Require:** weights $(w_i, c_i)_{i=1}^n$, size $K$, tolerance $\tau$
 1: Initialize: $\eta^{(0)} \leftarrow 0$
 2: **for** $k = 1, 2, \ldots, K$ **do**
 3:     $(S^{(k)}, \eta^{(k)}) \leftarrow \text{Alg1}\big((w, c), k, \tau, \eta^{(k-1)}\big)$
 4: **end for**
 5: **return** $(S^{(k)}, \eta^{(k)})_{k=1}^K$

---

**Computational Complexity** Each iteration of Algorithm 1 has three steps. First, the scores $s_i(\eta) = w_i + \eta c_i$ are computed in $\mathcal{O}(n)$ time. Second, the top-$k$ scores are selected, which can be done in expected $\mathcal{O}(n)$ time using linear-time selection, or in $\mathcal{O}(n \log k)$ time using a size-$k$ heap. Once the set is selected, $W(\mathbb{S})$ and $G(\mathbb{S})$ are computed in $\mathcal{O}(k)$ time. Thus, with linear-time selection, each iteration costs $\mathcal{O}(n)$ time and $\mathcal{O}(n)$ memory.

Let $I_k$ denote the number of Dinkelbach iterations for subset size $k$. The total cost for one size-$k$ problem is therefore $\mathcal{O}(I_k n)$. For an MIS path over $k = 1, \ldots, K$, the cost is $\mathcal{O}\left(n \sum_{k=1}^K I_k\right)$, with warm starts from Algorithm 2 typically reducing $I_k$. Because the feasible class is finite, the algorithm terminates finitely; empirically, only a small number of iterations is needed.

### 3.3. Optimization Exactness of Algorithm 1

The algorithm solves the fixed-input fractional optimization problem exactly. It converts the ratio problem into a sequence of linear maximization problems over a finite feasible class, each solved exactly by top-$k$ selection.

**Theorem 1** (Finite Exact MIS Selection). *Fix $n \in \mathbb{N}$, $k \in \{1, \ldots, n-1\}$ and assume Assumption 1 holds. Let $\eta^\star$ denote the optimal ratio value, let*

$$\mathcal{R}_k := \left\{ \frac{W(\mathbb{S})}{G(\mathbb{S})} : \mathbb{S} \subset [n], |\mathbb{S}| = k \right\},$$

*and set $M := |\mathcal{R}_k|$. Then for any $\eta^{(0)} \in \mathbb{R}$, Algorithm 1 terminates in at most $M+1$ ratio updates, and returns*

$$\mathbb{S}_k^{\max} \in \arg\max_{|\mathbb{S}|=k} \frac{W(\mathbb{S})}{G(\mathbb{S})}.$$

*Proof sketch.* Let $H(\eta) := \max_{|\mathbb{S}|=k} \{W(\mathbb{S}) - \eta G(\mathbb{S})\}$. Since $G(\mathbb{S}) > 0$, $H(\eta)$ has the sign of $\eta^\star - \eta$. At iteration $t$, the algorithm maximizes $H(\eta^{(t)})$ and updates to the ratio of the selected set. Hence updates from below $\eta^\star$ strictly increase the ratio, updates from above $\eta^\star$ move to a feasible ratio no larger than $\eta^\star$, and a zero residual is equivalent to optimality. Thus, after at most one update, the algorithm either terminates or moves strictly upward through distinct values in the finite set $\mathcal{R}_k$, terminating in at most $M$ ratio updates. See Section A1. □

Under absolute continuity and non-degeneracy of the realized weights and costs, ties between distinct feasible ratios occur with probability zero. In that case, $\hat{\mathbb{S}}_k^{\max}$ is almost surely unique.

### 3.4. Statistical Validity for Partial Linear Models

Algorithm 1 solves the fixed-input fractional optimization problem exactly. In a partial linear model, however, the inputs

$$w_i = \tilde{x}_i \tilde{r}_i, \qquad c_i = \tilde{x}_i^2$$

are generated by first-stage residualization. The statistical question is whether the empirical problem is close to the oracle problem based on the unknown residuals

$$v_i = x_i - h_0(Z_i), \qquad u_i = y_i - x_i \beta_0 - g_0(Z_i).$$

For fixed $k$, the relevant first-order oracle objective is additive. Let

$$\phi_i := \frac{v_i u_i}{\mu_v} \quad \text{with} \quad \mu_v := \mathbb{E}[v_i^2] > 0,$$

and define

$$Q_n^{\mathrm{or}}(\mathbb{S}) := \sum_{i \in \mathbb{S}} \phi_i, \qquad |\mathbb{S}| = k.$$

Let

$$\mathbb{S}_k^{\mathrm{or}} \in \arg\max_{|\mathbb{S}|=k} Q_n^{\mathrm{or}}(\mathbb{S})$$

denote an oracle first-order $k$-MIS. The empirical scaled objective is

$$\widehat{Q}_n(\mathbb{S}) := n\{\hat{\beta} - \hat{\beta}_{-\mathbb{S}}\} = n \frac{\sum_{i \in \mathbb{S}} \tilde{x}_i \tilde{r}_i}{\sum_{j \notin \mathbb{S}} \tilde{x}_j^2}.$$

For fixed $k$, define

$$E_{\mathbb{S}} := \sum_{i \in \mathbb{S}} (\tilde{x}_i \tilde{r}_i - v_i u_i), \qquad A_{\mathbb{S}} := \sum_{i \in \mathbb{S}} v_i u_i,$$

and

$$\delta_{n,k} := \sup_{\mathbb{S} \subset [n], |\mathbb{S}|=k} |E_{\mathbb{S}}|,$$

$$B_{n,k} := \sup_{\mathbb{S} \subset [n], |\mathbb{S}|=k} |A_{\mathbb{S}}|,$$

$$\rho_{n,k} := \sup_{\mathbb{S} \subset [n], |\mathbb{S}|=k} \left| \frac{1}{n} \sum_{j \notin \mathbb{S}} \tilde{x}_j^2 - \mu_v \right|.$$

**Assumption 2** (Uniform residualized-score stability). *For the fixed subset size $k$ under consideration,*

$$\rho_{n,k} = o_p(1), \qquad \delta_{n,k} = o_p(1), \qquad B_{n,k}\, \rho_{n,k} = o_p(1).$$

Primitive sufficient conditions include cross-fitting, bounded moments, non-degenerate treatment residual variance, and first-stage nuisance rates strong enough that the generated score error over fixed-size subsets is $o_p(1)$; details are given in the Appendix.

**Theorem 2** (First-order MIS validity). *Fix $k$ and suppose $\mu_v = \mathbb{E}[v_i^2] > 0$ and Assumption 2 holds. Then*

$$\sup_{\mathbb{S} \subset [n], |\mathbb{S}|=k} \left| \widehat{Q}_n(\mathbb{S}) - Q_n^{\mathrm{or}}(\mathbb{S}) \right| = o_p(1).$$

*Consequently, if*

$$\hat{\mathbb{S}}_k^{\max} \in \arg\max_{|\mathbb{S}|=k} \widehat{Q}_n(\mathbb{S}),$$

*then*

$$\left| \widehat{Q}_n(\hat{\mathbb{S}}_k^{\max}) - \max_{|\mathbb{S}|=k} Q_n^{\mathrm{or}}(\mathbb{S}) \right| = o_p(1).$$

*If the oracle first-order maximizer $\mathbb{S}_k^{\mathrm{or}}$ is unique and*

$$\Gamma_{n,k} := Q_n^{\mathrm{or}}(\mathbb{S}_k^{\mathrm{or}}) - \max_{\substack{|\mathbb{S}|=k \\ \mathbb{S} \neq \mathbb{S}_k^{\mathrm{or}}}} Q_n^{\mathrm{or}}(\mathbb{S}),$$

*satisfies*

$$\frac{\sup_{\mathbb{S} \subset [n], |\mathbb{S}|=k} \left| \widehat{Q}_n(\mathbb{S}) - Q_n^{\mathrm{or}}(\mathbb{S}) \right|}{\Gamma_{n,k}} = o_p(1),$$

*then*

$$\Pr(\hat{\mathbb{S}}_k^{\max} = \mathbb{S}_k^{\mathrm{or}}) \to 1.$$

*Proof sketch.* Write $D_{\mathbb{S}} = \sum_{j \notin \mathbb{S}} \tilde{x}_j^2$ and $d_{\mathbb{S}} = D_{\mathbb{S}}/n - \mu_v$. Then

$$\widehat{Q}_n(\mathbb{S}) = \frac{A_{\mathbb{S}} + E_{\mathbb{S}}}{\mu_v + d_{\mathbb{S}}}, \qquad Q_n^{\text{or}}(\mathbb{S}) = \frac{A_{\mathbb{S}}}{\mu_v}.$$

On the event $\rho_{n,k} < \mu_v/2$, uniformly over $|\mathbb{S}| = k$,

$$\left| \widehat{Q}_n(\mathbb{S}) - Q_n^{\text{or}}(\mathbb{S}) \right| \leq \frac{2}{\mu_v}|E_{\mathbb{S}}| + \frac{2}{\mu_v^2}|A_{\mathbb{S}}||d_{\mathbb{S}}|$$

$$\leq \frac{2}{\mu_v}\delta_{n,k} + \frac{2}{\mu_v^2}B_{n,k}\rho_{n,k}.$$

The right-hand side is $o_p(1)$ by Assumption 2. The value and selection claims follow from evaluating the uniform approximation at both the empirical and the oracle maximizer, using the gap condition for exact set recovery. See Section A2 for the full proof. $\qquad\square$

**Remark 3.1** (Scaling). *For fixed $k$, $\hat{\beta} - \hat{\beta}_{-\mathbb{S}} = O_p(n^{-1})$, so the meaningful first-order object is the scaled influence $n(\hat{\beta} - \hat{\beta}_{-\mathbb{S}})$. The limiting ranking is governed by the orthogonal scores $v_i u_i / \mathbb{E}[v_i^2]$.*

**Remark 3.2** (Value versus set recovery). *Value consistency is weaker and more stable than exact recovery of the selected set. If two oracle sets have nearly equal first-order values, small first-stage or sampling errors can change the identity of the selected set while leaving the maximum influence value essentially unchanged.*

**Remark 3.3** (Relation to orthogonal influence functions). *For fixed $k$, the PLM $k$-MIS is asymptotically governed by the orthogonal scores*

$$\phi_i = \frac{v_i u_i}{\mathbb{E}[v_i^2]}.$$

*The finite-sample objective, however, remains linear-fractional because the denominator $\sum_{j \notin \mathbb{S}} \tilde{x}_j^2$ varies with the removed set. Thus denominator effects, masking, and non-nestedness are finite-sample phenomena that vanish only at the first-order fixed-$k$ asymptotic scale.*

## 4. Experiments

We evaluate the method in simulations and applications. The simulations verify exact optimization against enumeration in small problems, and they measure runtime and iteration counts at scales where enumeration is impossible. The applications then use MIS traces as diagnostics in randomized experiments, linearized prediction tasks, and benchmark regression problems.

Across settings, we compute MIS in both signed directions, identifying sets that maximally increase and maximally decrease the target. We summarize results with *influence traces*, plotting the optimal influence value as a function of $k$. We also track non-nestedness events, $\hat{\mathbb{S}}_k \not\subset \hat{\mathbb{S}}_{k+1}$, and set overlap using the Jaccard index $J(\mathbb{A}, \mathbb{B}) := |\mathbb{A} \cap \mathbb{B}|/|\mathbb{A} \cup \mathbb{B}|$.

### 4.1. Simulations

We use simulations for two main purposes: (i) to validate theoretical predictions and (ii) to characterize runtime and iteration counts at scale.

**Optimization exactness** To validate Theorem 1, we compare Algorithm 1 with enumeration and greedy baselines in problems where exhaustive search is feasible ($n \leqslant 50$, $k \leqslant 3$). We use data-generating processes (DGPs) designed to stress the selection problem, including heteroskedasticity, heavy tails, mixtures, nonlinearities, endogeneity, autocorrelated errors, and masking. Across more than 10,000 Monte Carlo replications per design, Algorithm 1 always matches the enumerated optimum, while greedy selection sometimes returns suboptimal sets.

#### 4.1.1. RESIDUALIZED PERFORMANCE

We next assess the residualized setting motivated by Theorem 2. The simulation design contains two seeded influential groups of three observations each, built-in masking (following Kuschnig et al., 2021), and nonparametric confounding. We vary $n \in \{1000, 5000, 10000\}$, estimate the nuisance functions with gradient boosting and 5-fold cross-fitting, and compare the recovered empirical MIS with the known oracle-style score benchmark. Details are given in Appendix A4.

The residualized influence values track the oracle values closely, even at moderate sample sizes. Set recovery is more demanding; strongly influential sets are recovered reliably, whereas weakly separated or moderately influential sets converge more slowly. This matches Theorem 2 — value approximation is stable under weaker conditions than exact recovery of the maximizing set.

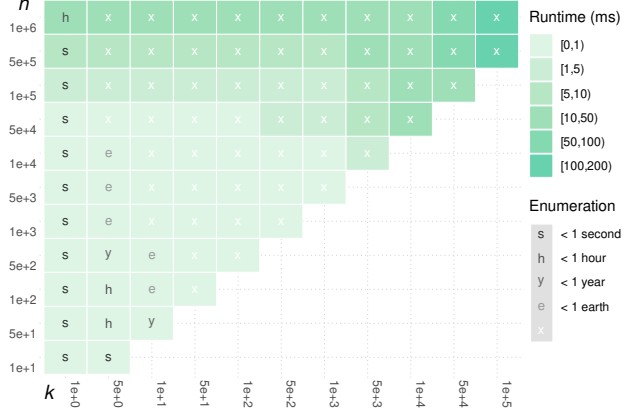

*Figure 2.* Runtimes of Algorithm 1 in milliseconds (median over 100 runs) for scenarios up to $n = 10^6$ and $k = 10^5$ (colored tiles), along with an enumeration feasibility frontier (markers) based on $\binom{n}{k}$ candidate sets under an optimistic per-set evaluation cost.

### 4.1.2. RUNTIME AND FEASIBILITY

We benchmark an implementation of Algorithm 1 on synthetic univariate regressions across a grid of sample sizes $n$ and set sizes $k$, recording wall-clock time (across 100 runs) and the number of Dinkelbach iterations. As shown in Figure 2, the median wall-clock time for $n = 10^6, k = 10^5$ remains below 200 ms.[3] Across the entire grid, Algorithm 1 converged in a median of three iterations, with a maximum of six.

To stress-test convergence, we further scale the implementation and use adversarial warm starts. At $n = 10^8$, runtime is roughly five seconds for $k = 10^6$ and 60 seconds for $k = 10^7$. At $n = 10^9$, where memory becomes limiting, runtime is about ten seconds for $k = 10^6$, 80 seconds for $k = 10^7$, and fifteen minutes for $k = 10^8$. For the $n = 10^8, k = 10^7$ case, replacing the warm start with $\eta^{(0)} \in \{10^3, 10^6, 10^9, 10^{12}, 10^{18}\}$ delays convergence by only one iteration, from five to six.

### 4.2. Applications

We use the method as a set-level sensitivity diagnostic in three classes of applications. In randomized experiments, the target is a fixed residualized univariate ratio and the algorithm identifies exact empirical MIS. In linearized prediction of semantic similarity, MIS quantify which training subsets most affect a local scalar prediction. In observational benchmark regressions, the PLM theory gives a first-order oracle interpretation when residualized-score stability is credible. In all applications we compute signed MIS in both directions.

### 4.2.1. RANDOMIZED CONTROLLED TRIALS

We revisit the seven microcredit RCTs studied by Meager (2019) and Broderick et al. (2023). These trials are useful stress tests because average treatment effect (ATE) estimates can be sensitive to small influential subsets — in two trials removing a single observation flips the sign of the estimated ATE, and sets of size 15 suffice across all trials. Previous work has used approximations, as enumeration is infeasible at the smallest size ($n \geqslant 961$). Our algorithm traces exact $k$-MIS over the full range of $k$ at negligible computational cost.

Figure 3 shows the Mongolia trial of Attanasio et al. (2015).

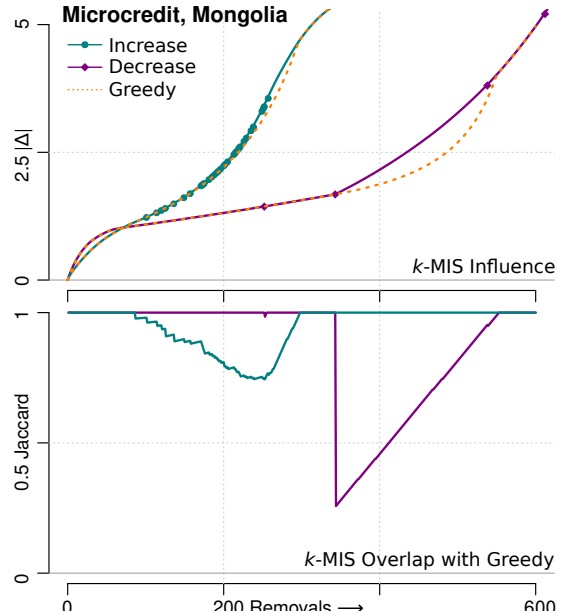

*Figure 3.* MIS impacts for a microcredit RCT conducted in Mongolia (Attanasio et al., 2015; Meager, 2019). Lines track $|\Delta_k|$ for the $k$-MIS that increase (teal) and decrease (purple) the ATE; the greedy approximation is overlaid as a dashed (orange) line. Points mark non-nestedness events. The bottom panel tracks the similarity of the $k$-MIS with its greedy approximation.

The increasing and decreasing traces differ substantially. In the increasing direction, non-nestedness is frequent but mostly local: the greedy trace remains close in value even as set overlap declines. In the decreasing direction, one non-nestedness event causes a large and persistent shift in the optimal set. Because greedy selection cannot revise earlier choices, it remains trapped on a suboptimal path and the influence gap accumulates.

Figure 4 reports the remaining six trials. Non-nestedness is common across trials, confirming that optimal MIS paths are generally not nested in $k$. Greedy selection often approximates influence magnitudes well, but it need not recover the maximizing sets. The Philippines trial features the clearest asymmetry — the increasing trace is initially stable and later accelerates, while the decreasing trace follows the opposite pattern. These results suggest that value approximation and set recovery should be evaluated separately, consistent with Theorem 2.

### 4.2.2. LINEARIZED TEXT EMBEDDINGS

Next, we use MIS for a linearized prediction task. We predict semantic similarity for $n = 5{,}749$ English sentence pairs from the STS benchmark (Cer et al., 2017). Each pair is mapped to frozen sentence embeddings using a pre-trained transformer model (Song et al., 2020), and the scalar regressor is the cosine similarity between the two em-

---

[3]The implementation uses a size-$k$ heap for ordering, and is implemented in R and Rcpp (Eddelbuettel & François, 2011), on a Ryzen AI Max+ Pro 395, using a single thread. We also benchmarked an implementation of the greedy algorithm by Kuschnig et al. (2021) on a subset of the grid (a runtime of 200 ms is achieved for $n = 10^4$ and $k = 50$), as well as the approximations implemented by Broderick et al. (2023); across a range of settings for $(n, k)$, our method is typically faster to compute, often by an order of magnitude.

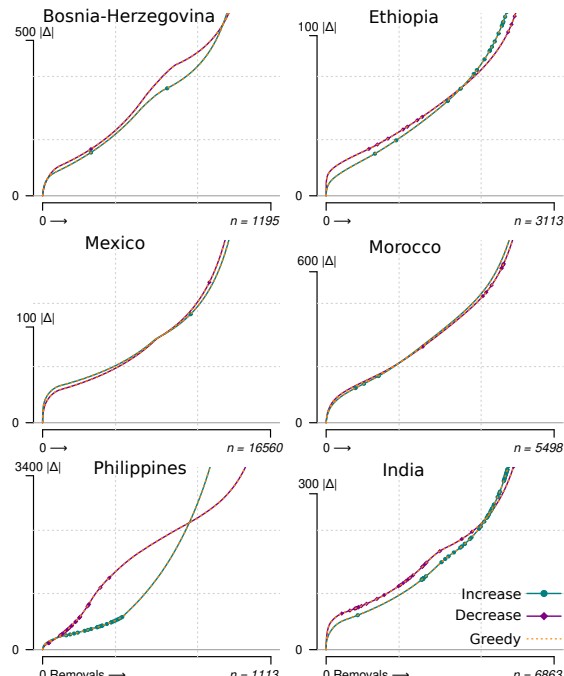

*Figure 4.* MIS impacts across the remaining six microcredit RCTs (see Meager, 2019). Lines show $|\Delta_k|$ for exact $k$-MIS (increase/decrease) and greedy; points mark non-nestedness events.

beddings. For three query pairs with low, medium, and high similarity, we study the local prediction target $\phi(\beta) = x_0\beta$.

*Table 1.* Sensitivity of fitted similarity scores to MIS.

| Query | $x_i$ | $y_i$ | $\hat{y}_{i_0}$ | Min | Max | $> 1$ |
|-------|-------|-------|-----------------|-----|-----|-------|
| Outside | 0.05 | 0.00 | 0.04 | 0.07 | 0.02 | – |
| ImClone | 0.67 | 0.54 | 0.58 | 1.02 | 0.30 | 4211 |
| Romney | 0.94 | 1.00 | 0.81 | 1.43 | 0.41 | 2545 |

Table 1 reports the full-sample fitted score, the fitted scores after removing the 5,000-MIS in each direction, and the smallest $k$ for which the fitted score exceeds 1.0 for three specific sentences (with minimal, average, and maximal similarity). Because the target scales with $x_0$, high-similarity query pairs are mechanically more sensitive to removal of influential training subsets. The traces reveal substantial sensitivity for the medium- and high-similarity pairs, including cases where deleting a subset pushes the linearized score outside the nominal $[0, 1]$ range. Further details are provided in Appendix A4.

### 4.2.3. ADDITIONAL APPLICATIONS

We also apply the method to benchmark datasets from economics, biology, statistics, and machine learning; details are in Appendix A4. Across these datasets, greedy selection usually tracks the optimal influence value closely, but

exact and greedy sets often differ. Most non-nestedness events are short-lived, in contrast to the persistent failure seen in the Mongolia trial. Warm-starting along the MIS path in Algorithm 2 keeps iteration counts small, with at most four iterations in these applications. Runtime is dominated by fitting the original model rather than by MIS selection.

## 5. Discussion

This paper makes most influential set analysis computationally explicit for a broad class of fixed-input univariate objectives. The key reduction is algebraic — once leave-set-out influence is written as a score-over-curvature ratio, the search over $\binom{n}{k}$ subsets reduces to repeated top-$k$ selection along a one-dimensional parameter path. For the fixed residualized inputs covered by Theorem 1, the result is the exact global optimum, not a greedy or local approximation.

The statistical interpretation depends on how those inputs are generated. In randomized experiments and other fixed-input settings, the empirical MIS is the target. In partial linear models, Theorem 2 shows that the scaled residualized objective has a first-order oracle interpretation when residualized scores and denominators are stable. This distinction is important: orthogonality supports value stability, but exact recovery of the selected set also requires separation between the best and second-best oracle sets.

### 5.1. Practical Use

We recommend reporting signed MIS traces over a range of $k$. The increasing and decreasing traces answer different robustness questions and can implicate different subsets. Large early jumps indicate sensitivity to few observations; gradual traces suggest dispersed influence; sharp composition changes signal non-nestedness and potential failure of greedy approximations. The implicated sets should then be inspected directly, especially when they contain duplicated records, unusual covariate patterns, high-leverage observations, or observations tied to a common source.

**Robustness Auditing** MIS traces provide target-specific worst-case deletion diagnostics. Compared with singleton influence measures, they reveal whether sensitivity is concentrated in one record or distributed across interacting observations. Compared with ad-hoc deletion checks, each set is the most adverse set of its size for the chosen target. Compared with greedy approximations, the exact trace is not constrained to follow a nested deletion path. Our experiments show that greedy methods may approximate the influence value while missing the maximizing set, and in some cases a single non-nestedness event produces persistent error.

**Data Curation** MIS can guide preprocessing and data cleaning without relying on blunt transformations such as trimming or winsorization. The selected sets identify which observations actually drive the downstream estimate or prediction. This can prioritize manual review, reveal duplicated or contaminated records, and document whether proposed cleaning rules address the relevant sources of sensitivity.

**Prediction Diagnostics** For linearized prediction targets, MIS provide dataset-level explanations that complement feature-level attributions. A large singleton effect indicates that one example matters on its own. A large set effect indicates that the prediction is anchored by a subset whose collective role may not be visible from individual records. This is useful for identifying duplicated, highly similar, or mutually masking examples that jointly support a prediction.

## 5.2. Limitations

The computational reduction requires a linear-fractional leave-set-out objective. This covers residualized univariate least-squares targets and related affine functionals, but non-linear estimands, vector-valued targets, and general $M$-estimators may require approximation, local linearization, or different optimization subproblems.

The PLM result is also deliberately first-order. For fixed $k$, the scaled residualized objective is close to an oracle orthogonal-score objective under residualized-score stability. This does not imply that arbitrary first-stage learners produce identical MIS, nor that exact set recovery is guaranteed without a gap. When first-stage learners are unstable for high-leverage or extreme-score observations, the recovered set remains the exact empirical MIS for the residualized regression but may not estimate the oracle first-order MIS.

Finally, exact MIS are diagnostic rather than prescriptive. A highly influential set may reveal contamination, model misspecification, subgroup heterogeneity, or genuine scientific signal. The method identifies the subset responsible for sensitivity; substantive interpretation still requires domain knowledge.

## 5.3. Extensions

Several extensions are natural. Ridge-stabilized denominators preserve the same basic reduction and can improve behavior when residual curvature is small. For broader classes of smooth estimators, approximate score-over-curvature representations may yield local or asymptotic analogues of the present algorithm. Structured versions of MIS are also promising; replacing the top-$k$ step with a constrained selection subproblem would allow influential clusters, time-contiguous blocks, group-balanced removals, or stratified deletion rules.

A second direction is inference on the traces themselves. Exact MIS computation makes it feasible to separate ordinary sampling variability from unusually influential subsets, and can provide candidate sets for formal tests of excessive influence (Konrad & Kuschnig, 2026). This shifts the next question from how to find influential sets to which influential sets are substantively or statistically anomalous.

## 5.4. Broader Implications

Efficient MIS computation turns set-level sensitivity analysis from a bespoke forensic exercise into a routine diagnostic. The method complements robust estimation, which limits the effect of contamination; meanwhile MIS identify which observations are responsible for sensitivity and how that sensitivity accumulates with set size. In high-stakes empirical and machine-learning pipelines, such inspectable counterfactual subsets can improve transparency, support data provenance, and clarify the relationship between data and conclusions.

## 6. Conclusion

We introduced an exact algorithm for identifying most influential sets when leave-set-out effects have a linear-fractional representation. The method reduces the combinatorial search over $\binom{n}{k}$ subsets to a finite sequence of top-$k$ selection problems, with $\mathcal{O}(n)$ cost per iteration and few iterations in practice.

The guarantees separate computation from statistical interpretation. For fixed residualized inputs, the algorithm returns a globally optimal empirical MIS. For partial linear models, the scaled residualized influence objective is first-order equivalent to an oracle orthogonal-score objective under residualized-score stability; value consistency follows directly, and exact set recovery follows under a separation condition.

Empirically, the method scales to datasets far beyond the reach of enumeration. Across randomized experiments, semantic similarity prediction, and benchmark regressions, exact MIS traces reveal frequent non-nestedness and show where greedy approximations succeed or fail. These traces provide actionable diagnostics for robustness auditing, data curation, and prediction analysis. Future work can extend the framework to structured deletion constraints, vector-valued or nonlinear targets, and formal inference for anomalous influence.

## Impact Statement

This paper develops methods for identifying most influential sets, whose removal induces the greatest change in a target quantity of interest. The primary goal is to improve robustness auditing and clarify how empirical conclusions depend on specific subsets of data.

Potential positive impacts include improved accountability, data provenance, and reliability. By identifying the subsets that drive estimates or predictions, the method can help practitioners detect contamination, duplicated records, unstable estimates, subgroup dependence, or other sources of sensitivity. It can also guide targeted data cleaning and make robustness analyses more reproducible.

A potential risk is adversarial use. An actor could use influence information to identify records whose inclusion or removal would steer a model or estimate in a desired direction. We view transparency as the appropriate mitigation. Set-level sensitivity already exists whether or not it is measured; efficient diagnostics make these dependencies visible, auditable, and easier to report.

The computational footprint is modest relative to enumeration. In our benchmarks, the method scales to millions of observations on a single thread, and the runtime is typically dominated by fitting the original model rather than by MIS selection.

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

# A1. Proof of Theorem 1

**Problem Formulation** Let $n \in \mathbb{N}$ and $k \in \{1, \ldots, n\}$. Given weights $w_i, c_i \in \mathbb{R}$ for $i = 1, \ldots, n$ and $T \in \mathbb{R}$ such that

$$T - \sum_{i \in \mathbb{S}} c_i > 0 \quad \text{for all } \mathbb{S} \subset [n] \text{ with } |\mathbb{S}| = k,$$

we consider the fractional programming problem:

$$\max_{\mathbb{S} \subset [n], |\mathbb{S}|=k} \frac{W(\mathbb{S})}{G(\mathbb{S})}, \qquad (A1.1)$$

where $W(\mathbb{S}) := \sum_{i \in \mathbb{S}} w_i$ and $G(\mathbb{S}) := T - \sum_{i \in \mathbb{S}} c_i$.

## Dinkelbach's Method

For $\eta \in \mathbb{R}$, define the parametric objective function:

$$F_\eta(\mathbb{S}) := W(\mathbb{S}) - \eta\, G(\mathbb{S}) = \sum_{i \in \mathbb{S}} (w_i + \eta c_i) - \eta T. \quad (A1.2)$$

For each $\eta \in \mathbb{R}$, define

$$z(\eta) := \max_{\mathbb{S} \subset [n], |\mathbb{S}|=k} \{W(\mathbb{S}) - \eta G(\mathbb{S})\}.$$

Because $G(\mathbb{S}) > 0$ on the feasible class, $\eta^\star = \max_{|\mathbb{S}|=k} \frac{W(\mathbb{S})}{G(\mathbb{S})}$ if and only if $z(\eta^\star) = 0$. Moreover, $z(\eta) > 0$ for $\eta < \eta^\star$ and $z(\eta) < 0$ for $\eta > \eta^\star$.

The key observation (Dinkelbach, 1967; Schaible, 1976) is that solving the fractional program equation A1.1 is equivalent to finding the parameter $\eta^*$ such that $z(\eta^*) = 0$.

**Theorem** (Finite Exact MIS Selection). *Fix $n \in \mathbb{N}$, $k \in \{1, \ldots, n-1\}$, and assume Assumption 1 holds. Let*

$$\eta^\star = \max_{|\mathbb{S}|=k} \frac{W(\mathbb{S})}{G(\mathbb{S})}$$

*and let*

$$\mathcal{R}_k = \left\{ \frac{W(\mathbb{S})}{G(\mathbb{S})} : \mathbb{S} \subset [n],\ |\mathbb{S}| = k \right\}.$$

*Set $M = |\mathcal{R}_k|$. With exact top-k selection and exact arithmetic, Algorithm 1 terminates finitely from any $\eta^{(0)} \in \mathbb{R}$ and returns*

$$\mathbb{S}_k^{\max} \in \arg\max_{|\mathbb{S}|=k} \frac{W(\mathbb{S})}{G(\mathbb{S})}.$$

*If $\eta^{(0)} \le \eta^\star$, termination occurs in at most $M$ ratio updates; for arbitrary $\eta^{(0)}$, termination occurs in at most $M + 1$ ratio updates.*

## Proof

**Lemma 1** (Distinctness of Ratio Values). *The number of distinct ratio values $N = |\{\Delta(\mathbb{S}) : \mathbb{S} \subset [n], |\mathbb{S}| = k\}|$ is finite with $N \le \binom{n}{k}$.*

*Proof.* Since there are $\binom{n}{k}$ selections and each has a well-defined ratio $\Delta(\mathbb{S}) = \frac{W(\mathbb{S})}{G(\mathbb{S})}$ (with $G(\mathbb{S}) > 0$), there are at most $\binom{n}{k}$ distinct ratio values. $\square$

**Lemma 2** (Monotone Progression Through Ratio Values). *The parameter sequence $\{\eta_\ell\}$ is monotone (strictly toward the optimum). Consequently, each distinct ratio value is visited by the parameter at most once. The algorithm does not cycle.*

*Proof.* By Dinkelbach's original analysis, the function $z(\eta)$ is continuous, convex, and strictly decreasing on intervals where the active maximizer has positive denominator. The update rule $\eta_{\ell+1} = \Delta(\mathbb{S}_\ell)$ ensures that the parameter moves toward the optimal value $\Delta_k^{\max}$ (see Schaible, 1976, for convergence rate analysis).

Consequently, the parameter sequence visits the distinct ratio values in a monotone order, each at most once. The algorithm cannot cycle because cycling would require revisiting the same parameter value, which contradicts monotonicity. $\square$

**Lemma 3** (Finiteness of Iterations). *The algorithm terminates in at most $N$ iterations, where $N$ is the number of distinct ratio values.*

*Proof.* By Lemma 2, the parameter sequence is monotone and visits each distinct ratio value at most once. By Lemma 1, there are finitely many distinct ratio values. Therefore, the algorithm terminates after at most $N$ iterations. $\square$

**Main Argument:** By Dinkelbach's fundamental result, the algorithm terminates when $z(\eta^*) = 0$, which certifies that $\eta^* = \Delta_k^{\max}$ and the corresponding selection is globally optimal. Lemmas 1–3 guarantee that this occurs in finite iterations. Thus, the algorithm outputs an exact global optimum. $\square$

**Remark A1.1** (Multiple Optima). *If multiple selections achieve the maximum ratio value $\Delta_k^{\max}$, the algorithm terminates upon reaching the first one. All selections with ratio $\Delta_k^{\max}$ are globally optimal.*

**Remark A1.2** (Practical Efficiency). *Although the worst-case bound is $O(N)$ where $N \le \binom{n}{k}$, empirical convergence is typically 3–10 iterations (Schaible, 1976) reflecting superlinear convergence in the parameter.*

**Remark A1.3** (Constructive Optimality). *Termination with $z(\eta_\tau) = 0$ provides a constructive certificate that $\mathbb{S}_\tau$ is globally optimal.*

## A2. Proof of Theorem 2

### Problem Formulation

Define

$$D_n(\mathbb{S}) := \sum_{j \notin \mathbb{S}} \tilde{x}_j^2, \qquad d_n(\mathbb{S}) := \frac{D_n(\mathbb{S})}{n} - \mu_v,$$

$$A_{\mathbb{S}} := \sum_{i \in \mathbb{S}} v_i u_i, \qquad E_{\mathbb{S}} := \sum_{i \in \mathbb{S}} (\tilde{x}_i \tilde{r}_i - v_i u_i),$$

so that $\sum_{i \in \mathbb{S}} \tilde{x}_i \tilde{r}_i = A_{\mathbb{S}} + E_{\mathbb{S}}$, and

$$\widehat{Q}_n(\mathbb{S}) = n(\hat{\beta} - \hat{\beta}_{-\mathbb{S}}) = \frac{A_{\mathbb{S}} + E_{\mathbb{S}}}{\mu_v + d_n(\mathbb{S})}, \qquad Q_n^{\mathrm{or}}(\mathbb{S}) = \frac{A_{\mathbb{S}}}{\mu_v}.$$

Write

$$\delta_{n,k} := \sup_{|\mathbb{S}|=k} |E_{\mathbb{S}}|, \ B_{n,k} := \sup_{|\mathbb{S}|=k} |A_{\mathbb{S}}|, \ \rho_{n,k} := \sup_{|\mathbb{S}|=k} |d_n(\mathbb{S})|,$$

all controlled by Assumption 2; in particular $\rho_{n,k} = o_p(1)$.

**Theorem** (First-order MIS validity). *Fix $k$ and suppose $\mu_v = \mathbb{E}[v_i^2] > 0$ and Assumption 2 holds. Then*

$$\sup_{\mathbb{S} \subset [n], |\mathbb{S}|=k} \left| \widehat{Q}_n(\mathbb{S}) - Q_n^{\mathrm{or}}(\mathbb{S}) \right| = o_p(1).$$

*Consequently, if*

$$\hat{\mathbb{S}}_k^{\max} \in \arg\max_{|\mathbb{S}|=k} \widehat{Q}_n(\mathbb{S}),$$

*then*

$$\left| \widehat{Q}_n(\hat{\mathbb{S}}_k^{\max}) - \max_{|\mathbb{S}|=k} Q_n^{\mathrm{or}}(\mathbb{S}) \right| = o_p(1).$$

*If the oracle first-order maximizer $\mathbb{S}_k^{\mathrm{or}}$ is unique and*

$$\Gamma_{n,k} := Q_n^{\mathrm{or}}(\mathbb{S}_k^{\mathrm{or}}) - \max_{\substack{|\mathbb{S}|=k \\ \mathbb{S} \neq \mathbb{S}_k^{\mathrm{or}}}} Q_n^{\mathrm{or}}(\mathbb{S}),$$

*satisfies*

$$\frac{\sup_{\mathbb{S} \subset [n], |\mathbb{S}|=k} \left| \widehat{Q}_n(\mathbb{S}) - Q_n^{\mathrm{or}}(\mathbb{S}) \right|}{\Gamma_{n,k}} = o_p(1),$$

*then*

$$\Pr(\hat{\mathbb{S}}_k^{\max} = \mathbb{S}_k^{\mathrm{or}}) \to 1.$$

### Proof

**Step 1: uniform approximation.** A direct rearrangement gives the identity

$$\widehat{Q}_n(\mathbb{S}) - Q_n^{\mathrm{or}}(\mathbb{S}) = \frac{E_{\mathbb{S}}}{\mu_v + d_n(\mathbb{S})} - \frac{A_{\mathbb{S}} d_n(\mathbb{S})}{\mu_v(\mu_v + d_n(\mathbb{S}))},$$

hence

$$\left| \widehat{Q}_n(\mathbb{S}) - Q_n^{\mathrm{or}}(\mathbb{S}) \right| \leq \frac{|E_{\mathbb{S}}|}{|\mu_v + d_n(\mathbb{S})|} + \frac{|A_{\mathbb{S}}| \, |d_n(\mathbb{S})|}{\mu_v \, |\mu_v + d_n(\mathbb{S})|}.$$

Since $\rho_{n,k} = o_p(1)$, the event $\{\inf_{|\mathbb{S}|=k} (\mu_v + d_n(\mathbb{S})) \geq \mu_v/2\}$ has probability tending to one. On this event, taking suprema yields

$$\sup_{|\mathbb{S}|=k} \left| \widehat{Q}_n(\mathbb{S}) - Q_n^{\mathrm{or}}(\mathbb{S}) \right| \leq \frac{2\,\delta_{n,k}}{\mu_v} + \frac{2\,B_{n,k}\,\rho_{n,k}}{\mu_v^2} = o_p(1).$$

**Step 2: value consistency.** Let

$$\hat{\mathbb{S}}_k \in \arg\max_{|\mathbb{S}|=k} \widehat{Q}_n(\mathbb{S}), \qquad \mathbb{S}_k^{\mathrm{or}} \in \arg\max_{|\mathbb{S}|=k} Q_n^{\mathrm{or}}(\mathbb{S}).$$

By optimality of $\hat{\mathbb{S}}_k$, $\widehat{Q}_n(\hat{\mathbb{S}}_k) \geq \widehat{Q}_n(\mathbb{S}_k^{\mathrm{or}})$, and combining this with the uniform approximation gives

$$Q_n^{\mathrm{or}}(\hat{\mathbb{S}}_k) \geq Q_n^{\mathrm{or}}(\mathbb{S}_k^{\mathrm{or}}) - 2 \sup_{|\mathbb{S}|=k} \left| \widehat{Q}_n(\mathbb{S}) - Q_n^{\mathrm{or}}(\mathbb{S}) \right|.$$

Hence the oracle value of the empirical maximizer is within $o_p(1)$ of the oracle optimum.

**Step 3: selection consistency.** Assume the oracle maximizer is unique, with gap

$$\Gamma_{n,k} := Q_n^{\mathrm{or}}(\mathbb{S}_k^{\mathrm{or}}) - \max_{\substack{|\mathbb{S}|=k \\ \mathbb{S} \neq \mathbb{S}_k^{\mathrm{or}}}} Q_n^{\mathrm{or}}(\mathbb{S}).$$

If $\hat{\mathbb{S}}_k \neq \mathbb{S}_k^{\mathrm{or}}$, then $Q_n^{\mathrm{or}}(\hat{\mathbb{S}}_k) \leq Q_n^{\mathrm{or}}(\mathbb{S}_k^{\mathrm{or}}) - \Gamma_{n,k}$, and Step 2 forces

$$2 \sup_{|\mathbb{S}|=k} \left| \widehat{Q}_n(\mathbb{S}) - Q_n^{\mathrm{or}}(\mathbb{S}) \right| \geq \Gamma_{n,k}.$$

Therefore, provided

$$\Gamma_{n,k}^{-1} \sup_{|\mathbb{S}|=k} \left| \widehat{Q}_n(\mathbb{S}) - Q_n^{\mathrm{or}}(\mathbb{S}) \right| = o_p(1), \qquad (\star)$$

we have $\Pr(\hat{\mathbb{S}}_k = \mathbb{S}_k^{\mathrm{or}}) \to 1$.

### A2.1. When does condition equation $\star$ hold?

Since $\rho_{n,k} = o_p(1)$ gives $\inf_{|\mathbb{S}|=k}\{\mu_v + d_n(\mathbb{S})\} = \mu_v + o_p(1)$, the bound of Step 1 sharpens to the rate

$$\sup_{|\mathbb{S}|=k} \left| \widehat{Q}_n(\mathbb{S}) - Q_n^{\mathrm{or}}(\mathbb{S}) \right| = O_p\left( \frac{\delta_{n,k}}{\mu_v} + \frac{B_{n,k}\,\rho_{n,k}}{\mu_v^2} \right).$$

Both summands being nonnegative, condition equation $\star$ is equivalent to

$$\Gamma_{n,k}^{-1}(\delta_{n,k} + B_{n,k}\,\rho_{n,k}) = o_p(1),$$

which in turn is equivalent to the two margin-dominance conditions holding jointly:

$$\boxed{\delta_{n,k} = o_p(\Gamma_{n,k}), \qquad B_{n,k}\,\rho_{n,k} = o_p(\Gamma_{n,k}).}$$

## A3. Joint Influence, Masking, and Greedy Failure

The following definitions are stated for a generic set objective

$$H_k(\mathbb{S}), \qquad \mathbb{S} \subset [n], \quad |\mathbb{S}| = k,$$

where, in the main applications,

$$H(\mathbb{S}) = \frac{W(\mathbb{S})}{G(\mathbb{S})} = \frac{\sum_{i \in \mathbb{S}} w_i}{T - \sum_{i \in \mathbb{S}} c_i}.$$

For signed analyses in the opposite direction, the same definitions apply after replacing $H$ by the corresponding direction-specific objective.

**Definition** (Marginal influence). *For a set $\mathbb{S} \subset [n]$ and an observation $i \notin \mathbb{S}$, the marginal gain from adding $i$ to $\mathbb{S}$ is*

$$\Delta(i \mid \mathbb{S}) := H(\mathbb{S} \cup \{i\}) - H(\mathbb{S}).$$

*When $\mathbb{S} = \varnothing$, we call $H(\{i\})$ the singleton influence of observation $i$.*

**Definition** (Joint influence). *For a set $\mathbb{S} \subset [n]$ with $|\mathbb{S}| = k$, define its excess joint influence relative to singleton effects as*

$$J(\mathbb{S}) := H_k(\mathbb{S}) - \sum_{i \in \mathbb{S}} H_1(\{i\}).$$

*The set $\mathbb{S}$ exhibits joint influence if $J(\mathbb{S}) > 0$, and it exhibits strong joint influence at level $\gamma > 0$ if*

$$H(\mathbb{S}) \geqslant \sum_{i \in \mathbb{S}} H_1(\{i\}) + \gamma.$$

*Thus joint influence occurs when the set effect is larger than what would be suggested by adding the singleton influences of its members.*

**Definition** (Masking). *Let $\mathbb{S}_k^\star \in \arg\max_{|\mathbb{S}|=k} H_k(\mathbb{S})$ be an optimal size-k set. An observation $i \in \mathbb{S}_k^\star$ is masked at size k if it belongs to an optimal size-k set but is not among the top k observations by singleton influence:*

$$i \in \mathbb{S}_k^\star \quad \text{and} \quad H_1(\{i\}) < H_1(\{j\}),$$

*for at least k observations $j \notin \mathbb{S}_k^\star$. Equivalently, singleton ranking alone would exclude i, even though i is part of an optimal influential set.*

**Definition** (Greedy path). *The greedy path is the sequence $(\mathbb{G}_k)_{k \geq 1}$ defined recursively by*

$$\mathbb{G}_0 = \varnothing, \quad \mathbb{G}_k = \mathbb{G}_{k-1} \cup \left\{ \arg\max_{i \notin \mathbb{G}_{k-1}} H_k(\mathbb{G}_{k-1} \cup \{i\}) \right\},$$

*with ties resolved by a fixed deterministic rule. The greedy path is nested by construction:*

$$\mathbb{G}_1 \subset \mathbb{G}_2 \subset \cdots .$$

**Definition** (Greedy failure). *Greedy selection fails at size k if the greedy set is not globally optimal:*

$$H_k(\mathbb{G}_k) < H_k(\mathbb{S}_k^\star).$$

*A sufficient structural reason for greedy failure is non-nestedness; if an early greedy choice is absent from every optimal size-k set, then the nested greedy path cannot recover the optimal size-k solution.*

For the linear-fractional objective in the main text, marginal gains depend on the current set. Writing $w_i = \tilde{x}_i \tilde{r}_i$, $c_i = \tilde{x}_i^2$, $W(\mathbb{S}) = \sum_{j \in \mathbb{S}} w_j$, and $G(\mathbb{S}) = T - \sum_{j \in \mathbb{S}} c_j$, we have

$$H(\mathbb{S} \cup \{i\}) - H(\mathbb{S}) = \frac{w_i G(\mathbb{S}) + c_i W(\mathbb{S})}{G(\mathbb{S})\{G(\mathbb{S}) - c_i\}}.$$

Thus the gain from adding $i$ depends not only on its own score $w_i$ and curvature $c_i$, but also on the accumulated score $W(\mathbb{S})$ and remaining curvature $G(\mathbb{S})$ of the current set. This set dependence allows joint influence, masking, non-nested optimal paths, and greedy failure.

## A4. Additional Results

### Influence Convergence Simulation

We consider the partial linear model from Section 2. To evaluate the finite sample behavior of Theorem 2, we generate data with the following structure. Covariates are drawn as $Z_{ij} \overset{\text{iid}}{\sim} \text{Uniform}(0,1)$ for $j = 1, \ldots, p_z$. Treatment assignment incorporates confounding through

$$X_i = h(\mathbf{Z}_i) + v_i, \tag{A4.3}$$

where $h(Z) = 0.5z_1 - 0.2z_2$ and $v_i$ follows a mixture distribution to create high leverage sets:

$$v_i \sim \begin{cases} \mathcal{N}(8, 0.25^2) & \text{for } i = n, n-1, n-2, \\ \mathcal{N}(6, 0.25^2) & \text{for } i = n-3, n-4, n-5, \\ \mathcal{N}(0, 1) & \text{for } 1, \ldots, n-6. \end{cases}$$
$$\tag{A4.4}$$

The nonparametric component in $Y$ exhibits sparsity, depending on only four covariates:

$$g(\mathbf{z}) = 2\sin(2\pi z_1) + 1.5z_2^2 + z_3 z_4. \tag{A4.5}$$

**Heterogeneous Treatment Effects.** To ensure high influence for some observations, we generate outcomes with treatment effect heterogeneity, violating constant $\beta$:

$$Y_i = g(\mathbf{Z}_i) + \beta_i X_i + \varepsilon_i, \tag{A4.6}$$

where

$$(\beta_i, \varepsilon_i) \sim \begin{cases} (-0.5, \mathcal{N}(0, 1)) & \text{for } i = 1, \ldots, n-6 \\ (0.1, \mathcal{N}(0, 0.1^2)) & \text{for } i = n-5, n-4, n-3 \\ (0.4, \mathcal{N}(0, 0.1^2)) & \text{for } i = n-2, n-1, n. \end{cases}$$

This design extends the illustration of masking and joint influence in Kuschnig et al. (2021). Each setup is resampled, for a total of one thousand Monte Carlo draws. Figure A1 displays the mean and 95% confidence bounds of the simulation results.

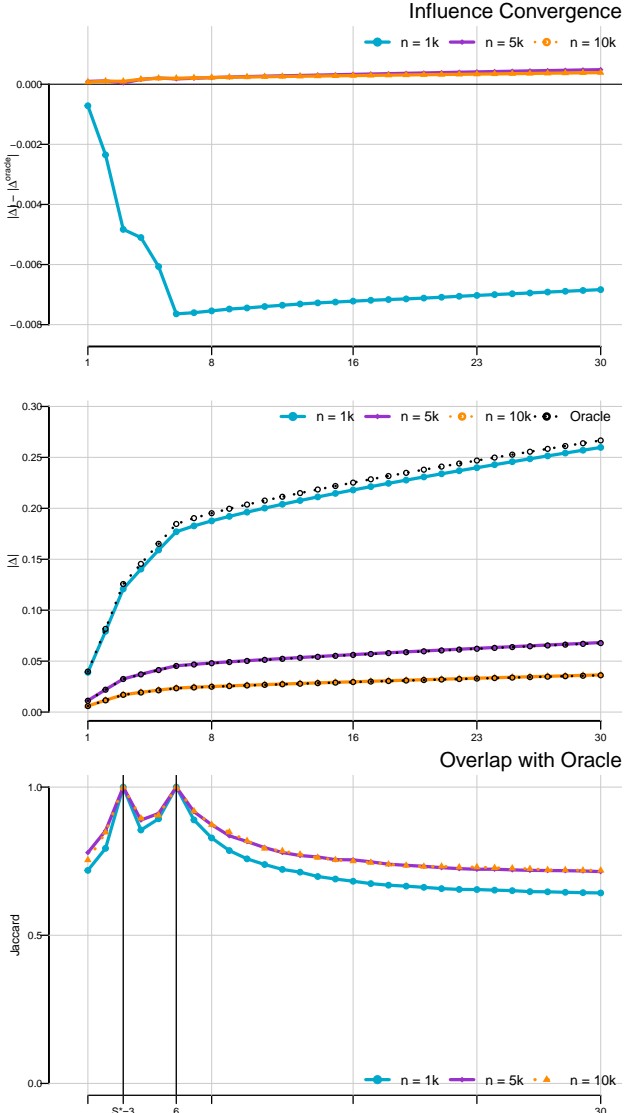

*Figure A1.* Simulation study to assess finite sample convergence of the limiting results. Specifically, $\Delta(\hat{\mathbb{S}}_k) \to \Delta^{\text{oracle}}(\mathbb{S}_k^{\max})$ (top) and $\hat{\mathbb{S}}_k \to \mathbb{S}_k^{\text{oracle}}$ (bottom). Per construction $k = 3$ and $k = 6$ are particularly influential.

**Simulation Results**  While influence converges quickly and uniformly to the oracle, the associated $\mathbb{S}^\star$ converges much slower. While even for medium sized sample size residualization works well with an average Jaccard similarity of almost one for all sample sizes, accuracy is substantially worse if added observations are not influential. Thus knowing the correct $k$ is beneficial for set accuracy in finite

samples. If inference on $\mathbb{S}^\star$ is not desired but merely robustness, even for moderate $n$ residualization recovers the true $\Delta_k^{\max}$ well.

### A4.1. Linearized Text Embeddings

The sentence-pairs and similarity scores are:

100% 'Romney picks Ryan as vice presidential running mate: source'

100% 'Romney to tap Ryan as vice presidential running mate'

0% 'An older man is standing outside in front of a truck.'

0% 'A woman dressed in green is roller skating outside at an event.'

54% 'A new study, conducted in Europe, found the medicine worked just as well as an earlier disputed study, sponsored by ImClone Systems, said it did.'

54% 'Doctors concluded Erbitux, the cancer drug that enmeshed ImClone Systems in an insider trading scandal, worked just as well as an earlier company-sponsored study said it did.'

Traces of the predicted similarity scores as $k$ increases are shown in Figure A2.

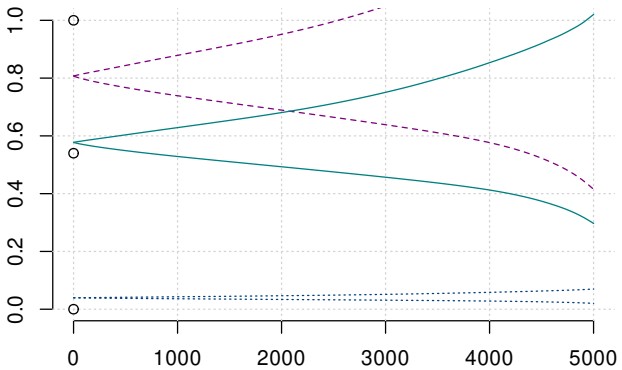

*Figure A2.* MIS traces for three sentence predictions. Points depict true values.

### A4.2. Additional Applications

The datasets considered are listed in Table A1; results are provided as Supplementary Material.

### A4.3. Comparison with Influence Function Approximations

Figure A3 compares exact $\Delta(\mathbb{S})$ with their AMIP approximation (Broderick et al., 2023).

*Table A1.* Analyzed datasets, sample sizes, and sources.

| Name | $n$ | Source |
|---|---|---|
| Saltmarsh Sparrow | 1295 | Gjerdrum et al. (2008) |
| Mistrust | 20062 | Nunn & Wantchekon (2011) |
| Tsetse Fly | 522 | Alsan (2015) |
| Wine Quality | 1599 | Cortez et al. (2009) |
| Adult Income | 32561 | Kohavi (1996) |
| California Housing | 20640 | Pace & Barry (1997) |
| Boston Housing | 506 | Harrison & Rubinfeld (1978) |
| Diamonds | 53940 | Wickham (2016) |
| Star Cluster | 47 | (Maechler et al., 2025) |
| AM Example | 12 | (Maechler et al., 2025) |
| Brain to Body | 65 | (Maechler et al., 2025) |
| Lactic Acid | 20 | (Maechler et al., 2025) |
| Plant Data | 20 | (Maechler et al., 2025) |
| Telephone Calls | 24 | (Maechler et al., 2025) |
| Stackloss | 21 | (Maechler et al., 2025) |
| HBK Data | 75 | (Maechler et al., 2025) |
| Wood | 20 | (Maechler et al., 2025) |
| Salinity | 28 | (Maechler et al., 2025) |
| Phosphorus | 18 | (Maechler et al., 2025) |
| Education | 50 | (Maechler et al., 2025) |
| Aircraft Data | 23 | (Maechler et al., 2025) |
| May Air Pollution | 31 | (Maechler et al., 2025) |
| Bushfire | 38 | (Maechler et al., 2025) |
| Cloud Point | 19 | (Maechler et al., 2025) |
| Soil Chemistry | 428 | (Maechler et al., 2025) |
| Heart Catheter | 12 | (Maechler et al., 2025) |
| Pension Fund | 18 | (Maechler et al., 2025) |
| Pulp Fiber | 62 | (Maechler et al., 2025) |
| Toxicity | 38 | (Maechler et al., 2025) |
| Wagner Growth | 63 | (Maechler et al., 2025) |

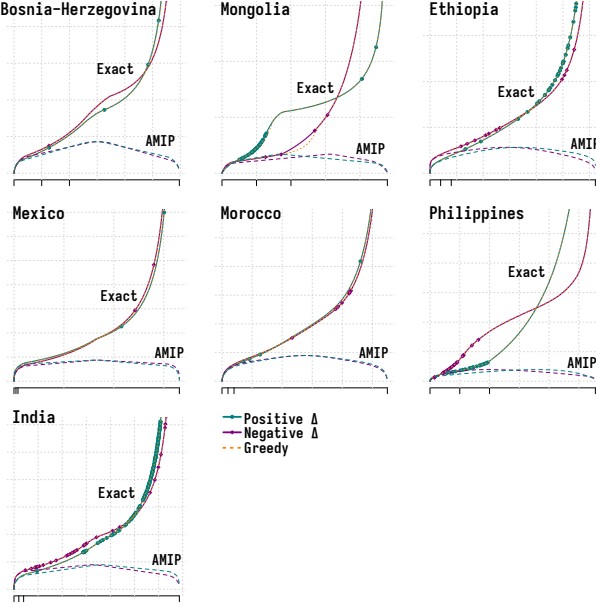

*Figure A3.* Comparison of exact MIS impacts on the seven microcredit trials with their approximate maximum influence perturbation (Broderick et al., 2023).

