# OpenReview forum: "Finding Most Influential Sets"
_ICML.cc/2026/Conference — ICML 2026 regular_

### Official Review · Reviewer_EEDE · 2026-03-02

**Soundness:** 3
**Presentation:** 3
**Significance:** 2
**Originality:** 3
**Overall Recommendation:** 5
**Confidence:** 3

**Summary:**

This paper introduces an efficient algorithmic framework for identifying most influential sets. Traditionally, finding these sets was considered a computationally impossible combinatorial problem requiring a search over $\binom{n}{k}$ subsets. The authors demonstrate that for a wide class of leave-set-out effects, the problem can be reformulated as a linear-fractional program. The authors then use Dinkelbach's method to build a MIS searching algorithm, whose complexity is O(n). The authors provide theoretical guarantees of global optimality for univariate settings and selection consistency for partial linear models under Neyman orthogonality. Empirically, the algorithm avoids the suboptimal local maxima common in greedy heuristics and can process datasets of $n=10^6$ in under 200 milliseconds, making exact set-level sensitivity analysis a routine diagnostic for modern machine learning and statistical pipelines.

**Compliance With Llm Reviewing Policy:**

Affirmed.

**Final Justification:**

The rebuttal has addressed all of my concerns. My recommendation on the paper is accept.

**Key Questions For Authors:**

1. Could the authors briefly state the proof sketch of Proposition 1 in the main body or at least in the appendix? This is the core conclusion for the proposed algorithm to work. A brief proof will be helpful for interested readers.

2. In Theorem 1, M is the number of different ratio values. Will this quantity be of order $n^2$ or $n$? If the former holds, why is this different from the claim of linear convergence in line 182? If the latter holds, how is $M=O(n)$ achieved?

**Limitations:**

See weaknesses and questions.

**Strengths And Weaknesses:**

Strengths:

1. The authors utilize a technical equivalence form of the difference of treatment effect estimators to build a simple yet efficient characterization of the target function. It is then integrated with a linear time optimization method to build a MIS searching algorithm. The propoed algorithm is a skillful integration of math and optimization building blocks.

2. The authors prove that the output of the proposed algorithm is consistent under certain mild assumptions.

3. The paper's overall presentation is good. The introduction part is logically coherent with a clear explanation of why the problem is important, how existing methods fail to provide acceptable solutions and what the paper will propose. It is clear even without math expressions.

Weaknesses:

1. The model is restricted to partial linear models, which could be prohibitive in more general scenarios.

2. The problem setting is restricted to univariate coefficient estimation. The MIS problem could be important in many other problems like parameter estimation or model fitting.

---

> ### Author Rebuttal · Authors · 2026-03-31
>
> We thank the reviewer for the positive assessment of the paper's technical contribution, clarity, and overall presentation. We are encouraged that the formulation and algorithmic integration are viewed as effective and well-motivated.
>
> ### Scope of the model
>
> We agree that the scope is restricted in two important ways: (1) the exact optimization result is developed for univariate targets, and (2) the statistical consistency result is established for partial linear models.
>
> These restrictions are meaningful, but deliberate.
> They allow us to isolate the main optimization and statistical ideas in a *clear, accessible, and practically relevant setting* without adding technical overhead that would obscure the core contribution.
> At the same time, we agree that *extensions to multivariate targets and broader model classes* are important directions for future work.
> In our view, the current form of the paper facilitates such extensions by establishing a reusable algorithmic primitive to be embedded in richer settings, while already covering a widely used class through the double-ML framework.
>
> ### Proof of Proposition 1
>
> We agree that a proof sketch should be included.
> We will add a short, self-contained derivation in the main text to make the steps explicit and easy to verify. We will also include a brief explanation of the resulting score–curvature decomposition that enables the linear-fractional reformulation.
>
> A sketch is as follows:
>
> #### Proof sketch
>
>   Rearrange the full-sample normal equation over observations not in $\mathbb{S}$:
>   $$
>       \sum_{i\notin \mathbb{S}} \tilde x_i \tilde y_i =
>           \hat\beta \sum_{i\notin \mathbb{S}} \tilde x_i^2 +
>               \sum_{s\in \mathbb{S}}\tilde x_s(\hat\beta \tilde x_s-\tilde y_s).
>   $$
>   Using $\tilde r_s = \tilde y_s - \hat\beta \tilde x_s$, this becomes
>   $$
>       \sum_{i\notin \mathbb{S}} \tilde x_i \tilde y_i =
>           \hat\beta \sum_{i\notin \mathbb{S}} \tilde x_i^2 -
>               \sum_{s\in \mathbb{S}}\tilde x_s \tilde r_s.
>   $$
>   Divide both sides by $\sum_{i\notin \mathbb{S}}\tilde x_i^2$ to obtain
>   $$
>       \hat\beta_{-\mathbb{S}} =
>           \hat\beta -
>               \frac{\sum_{s\in \mathbb{S}}\tilde x_s \tilde r_s}{\sum_{i\notin \mathbb{S}}\tilde x_i^2},
>   $$
>   which is the claimed result.
>
> ### Theorem 1 and $M$
>
> We agree that Theorem 1 and the role of $M$ require clarification.
>
> The theorem is a finite-termination statement, where $M$ is a worst-case upper bound on the number of distinct ratio values encountered by the algorithm. This is a combinatorial bound, $M \leq \binom{N}{k}$.
> This is distinct from (a) the *per-iteration* computational cost (an $O(n)$ selection), and (b) the *convergence-rate* statement in line 182.
> The algorithm converges in few iterations (empirically in the single digits), even though the objective admits many candidate ratios.
>
> This fast convergence arises because each iteration solves the parametric subproblem exactly and updates the induced ratio. As a result, the method acts as an exact root-finding procedure for the envelope function, rather than a local search over subsets. The iterates move monotonically toward the optimum.
> Intuitively, this resembles a Newton-style update, where each step makes a targeted, global correction rather than a local improvement.
>
> The worst-case bound $M \leq \binom{n}{k}$ is not approached in practice.
> The envelope function $f(\eta) = \max_{\mathcal{S}} F_\eta(\mathcal{S})$ is convex and piecewise linear in $\eta$, and each Dinkelbach update moves directly to the current maximizer ratio. As a result, parameter convergence to $\eta^\star = \Delta_k^{\max}$ is superlinear despite the combinatorial feasible set.
>
> We will revise the text to clearly separate (1) finite-termination, (2) per-iteration complexity, and (3) convergence rate.
>
> ### Summary
>
> We will:
>
> - clarify the scope and discuss concrete extensions,
> - include a proof sketch of Proposition 1 with supporting intuition, and
> - clarify the role of $M$ and distinguish finite-termination, complexity, and convergence.
>
> We thank the reviewer for their helpful suggestions, which will improve clarity and precision of the manuscript.

---

> > ### Author Rebuttal · Reviewer_EEDE · 2026-04-01
> >
> > I appreciate the authors' detailed responses and clarifications. I have updated by score.

---

### Official Review · Reviewer_hwRS · 2026-03-09

**Soundness:** 4
**Presentation:** 4
**Significance:** 3
**Originality:** 4
**Overall Recommendation:** 5
**Confidence:** 4

**Summary:**

The paper addresses the problem of identifying “most influential sets” (MIS) of training examples – subsets whose removal causes the greatest change in a target model’s output or estimate. To tackle this combinatorially hard problem (selecting top-$k$ influential points from $n$), the authors reformulate the influence measure in a linear-fractional form and apply Dinkelbach’s algorithm (a classic method for fractional optimization). This yields an efficient procedure (provably convergent in finite steps) that finds the optimal influential subset without brute force search. The paper provides theoretical guarantees under certain conditions (e.g., proving global optimality in simplified cases and statistical consistency in a partial linear model) and conducts extensive experiments on both synthetic and real datasets. Empirically, the proposed method consistently identifies more truly influential subsets than baseline heuristics (like greedy algorithms) and does so with dramatically better scalability.

**Compliance With Llm Reviewing Policy:**

Affirmed.

**Key Questions For Authors:**

no

**Limitations:**

yes

**Strengths And Weaknesses:**

**Soundness:** The submission is technically solid and rigorous. The core algorithmic idea (reducing the leave-$k$-out influence search to a one-dimensional continuous optimization) is logically derived and supported by theory. The authors clearly describe Algorithm 1 implementing Dinkelbach’s method and prove its finite convergence to the optimal subset under the given formulation. The theoretical analysis (e.g., Theorem 2 for consistency in partial linear models) is grounded in standard statistical assumptions (such as Neyman orthogonality and stable nuisance estimators), and the proofs appear correct. The experimental methodology is thorough: the paper verifies correctness on small cases by comparing to exhaustive search, and evaluates performance on larger tasks with appropriate metrics. All claims are backed by either theoretical arguments or empirical evidence. The main caveat is that the theoretical guarantees rely on fairly strong conditions. If these assumptions are violated (for instance, in highly complex settings) the method might flag points that are “influential” due to modeling artifacts or noise rather than true influence. The authors do acknowledge this limitation, noting that the MIS identified may reflect instability in the first-stage model when conditions fail. This is not a fundamental flaw in correctness (the algorithm still does what it is designed to do), but it means practitioners should be cautious to ensure conditions approximately hold.

**Presentation:** The paper is clearly written and well-structured, meeting the standards of a top-tier conference submission. The introduction motivates the problem of influential data subsets and situates it well in the context of prior work (e.g., limitations of influence functions and greedy approaches). The methodology section is detailed: it presents the problem setup, the fractional reformulation, and the pseudocode for the algorithm in a logical progression. Key theoretical results are stated clearly (with proofs in the appendix), and the authors include an illustrative toy example that helps build intuition about why a greedy selection can fail. The experimental section is comprehensive and supported by informative graphs, which help the reader see the performance differences between methods. Overall, the narrative is easy to follow. Concepts like fractional programming and Dinkelbach’s algorithm are briefly described, and there are references to further reading for unfamiliar readers. The writing is professional and precise, with claims and methodology explained at an appropriate level of detail. A minor aspect is that some sections assume background knowledge that not all ICML readers may have. For example, terms from causal inference and advanced estimation theory (e.g., “Neyman orthogonality”) appear with relatively brief explanation. Likewise, the use of Dinkelbach’s method, while central, could be given a bit more intuitive description in the main text for readers new to fractional optimization. A small typo was likely identified: In Section 2.1, the assumption $\mathbb{E}[u_i^2 \mid Z_i] = \sigma^2$ is stated. Should this not be  $\mathbb{E}[u_i^2 \mid Z_i, x_i] = \sigma^2$?

**Significance:**  This work addresses an important and previously elusive problem in machine learning - identifying influential groups of training data. This is significant because understanding data influence at the subset level can enhance model robustness, facilitate data cleansing, and improve interpretability (e.g., detecting sets of examples that collectively bias a model’s predictions). Prior to this work, the combinatorial nature of the problem meant that researchers mainly relied on surrogates or approximate methods for group influence (influence functions, leave-one-out heuristics, or worst-case bounds). By providing a tractable algorithm for exact top-$k$ subset influence under certain conditions, the paper potentially opens up new possibilities for practitioners and researchers.

**Originality:** The paper’s key idea is quite original. Identifying influential subsets has been an open challenge, and the authors’ approach is novel within this context. As far as I am aware, no prior work has used fractional programming (via Dinkelbach’s algorithm or similar) to exactly solve the top-$k$ influence subset identification problem. Previous approaches typically relied on greedy additive strategies or approximations that do not guarantee finding the truly most influential set. By contrast, this work shows a clever way to transform the discrete selection problem into a continuous one where global optima can be found efficiently. This is a non-trivial and creative contribution. The theoretical guarantees (finite-step convergence to the optimum for fixed design, statistical consistency under certain models) further distinguish this method from past heuristic attempts.

---

> ### Author Rebuttal · Authors · 2026-03-31
>
> We thank the reviewer for their careful reading and positive assessment of our paper's soundness, presentation, significance, and originality.
> We are encouraged that the core algorithmic idea is viewed as technically solid, the experiments as thorough, and the contribution as novel and important.
>
> We appreciate the suggestions to improve presentation and agree that the current draft can better expose key technical ingredients to readers without specialized background.
> In the revision, we will improve the exposition in three main ways:
>
> 1. We will add intuition for *Neyman orthogonality* and its role in Theorem 2, clarifying that orthogonality ensures the residualized objective tracks its oracle counterpart
> 2. We will expand the discussion of *joint influence, masking, and greedy search*. While the current example points in this direction, these ideas will be treated more explicitly in the main text.
> 3. We will provide additional intuition for the *score-curvature ratio* implied by Proposition 1, and explain how Dinkelbach updates exploit this structure. We will include a concise proof of Proposition 1, and clarify the connection between the deletion identity, the linear-fractional objective, and the resulting optimization algorithm.
>
> We thank the reviewer for identifying the typo in Section 2.1; the conditioning statement has been corrected.
>
> We believe these revisions will improve clarity and accessibility while strengthening the presentation of the paper's contributions.

---

> > ### Author Rebuttal · Reviewer_hwRS · 2026-04-01
> >
> > Thank you, my concerns have been fully addressed.

---

### Official Review · Reviewer_Ncj2 · 2026-03-10

**Soundness:** 3
**Presentation:** 2
**Significance:** 2
**Originality:** 2
**Overall Recommendation:** 2
**Confidence:** 3

**Summary:**

This paper studies the identification of most influential sets (MIS), defined as size-k subsets whose removal maximally changes a target estimand. In general, this problem is combinatorial and computationally intractable. The authors show that under a partial linear model assumption, the MIS objective can be reformulated as a linear-fractional optimization problem. Using Dinkelbach’s method, the problem reduces to a sequence of top-k selection steps, resulting in a linear-time algorithm.

**Compliance With Llm Reviewing Policy:**

Affirmed.

**Final Justification:**

The authors kind of avoided my question on the greedy baseline during the rebuttal. To respond to my point of weak evaluation, they added an additional baseline that is weaker than greedy in PLM. Overall, I do not find authors argument convincing.

I agree with the authors that there are benefits to knowing the exact influential set. However, if we judge the paper as a theory paper, as the authors requested, then the analysis techniques are standard. The paper also spent one and a half page discussing their results and recommendations, which is unusual for a theory-focused paper. If we judge the paper as an applied paper, then the greedy algorithm again performs quite well in the PLM setting. So I don't really see the need of developing such an algorithm.

Combining these, I don't recommend accepting the paper.

**Key Questions For Authors:**

Please address the questions above. The main concern is with the limited benchmark comparisons in the evaluation strategy.

**Limitations:**

yes

**Strengths And Weaknesses:**

**Soundness**


- Empirical evaluation is limited: The experiments compare only to greedy and enumeration (the latter only for tiny $n$). No comparisons to influence functions, group influence methods, or Shapley-based approaches are provided. The paper's own figures (Figures 3 and 4) show that greedy tracks influence magnitude well most of the time; the claimed advantage is in exact set composition, but the practical value of exact set recovery is not demonstrated.

- No validation of interpretability claims: Despite asserting that exact MIS enables interpretability and data auditing, the paper provides no qualitative analysis of the identified points. It does not show that exact sets contain more actual data errors, reveal meaningful patterns, or lead to different conclusions than greedy approximations.

**Presentation**

- Narrative overstates contribution: The paper frames the problem as an intractable combinatorial search ({n} choose {k}) and presents the method as a breakthrough. However, the problem is already simplified by the PLM structure, which yields a closed-form ratio. The narrative is compelling but mismatched to the modest technical advance.

- Discussion section overreaches: Nearly two pages are devoted to speculative implications (fairness, accountability, transparency, data provenance) without evidence that the method delivers on these fronts. This inflates the perceived importance of the work beyond its demonstrated scope.

- Missing benchmarks: The authors only compare greedy benchmarks and do not compare against additional methods, such as existing influence function-based methods.

- Clarity is good but selective: The exposition of the algorithm and theory is clear, but the framing obscures the strong assumptions required. The claim that the method is ``exact in RCTs'' glosses over the fact that this exactness depends on using a specific estimator (residualized OLS), not all estimators commonly used in practice. The introduction starts with the general goal of influential sample selection in ML, yet Section 2 focuses on the PLM.


**Significance**

- Model class is highly restrictive: The PLM with univariate treatment, additive noise, and constant effects is a narrow subclass of models. For modern ML applications (deep learning, LLMs, heterogeneous effects), the method does not apply. The paper's significance is thus limited to a specific audience (applied economists, epidemiologists using linear adjustment).

- Value of exact set recovery is unsubstantiated: The paper argues that exact MIS matters for interpretability and auditing, but provides no evidence that examining the exact set yields insights beyond examining a greedy approximation with similar influence magnitude. Without qualitative validation, the claim remains speculative.

- Relevance to ML interpretability is overstated: The machine learning example (linearized text embeddings) is essentially a linear model on pre-trained features. This does not demonstrate applicability to the non-convex, high-dimensional models that dominate modern ML.

- Small-sample auditing is not addressed: Asymptotic guarantees do not help in the small-sample regimes where auditing is most critical. The paper does not provide finite-sample results or guidance for practitioners on when to trust the method.


**Originality**

- Core ratio result is not original: Proposition 1, which underpins the entire method, is taken directly from Konrad \& Kuschnig (2025). The paper's contribution is recognizing this as a linear-fractional program and applying Dinkelbach's method.

- Optimization approach is standard: Dinkelbach's algorithm for fractional programming (1967) and its application to top-$k$ selection with scores $w_i + \eta c_i$ is mathematically straightforward. The reduction is elegant but not technically deep.

- Statistical theory follows the existing DML framework: The consistency argument (Theorem 2) is a direct application of Chernozhukov et al. (2018)'s DML theory to set selection. No new statistical principles are introduced.

---

> ### Author Rebuttal · Authors · 2026-03-31
>
> We thank the reviewer for their detailed report. We address their concerns below and will revise the paper accordingly.
>
> ### Clarification of contributions
> The paper makes two distinct contributions that we will separate more clearly:
>
> 1. an **exact finite-sample optimization result** for MIS in the univariate setting, and
> 2. a **consistency result** for MIS under standard double/debiased ML assumptions.
>
> ## Empirical evaluation and benchmarks
>
> We understand the reviewer's concern regarding limited benchmarks, and will expand this aspect of the paper.
>
> We will add comparisons to the AMIP method of Broderick, Giordano, and Meager (2021) using their `zaminfluence` package across the seven microcredit RCTs. (Results omitted due to space constraints)
>
> - Preliminary results show substantial improvements in runtime and objective value of the identified MIS.
> - We will clarify that our method computes the *exact finite-sample optimum*, whereas influence methods approximate marginal effects. These approaches are complementary rather than directly comparable, but the comparison is still informative and will be included.
>
> Regarding existing experiments:
>
> - Enumeration is included precisely to demonstrate **infeasibility beyond** small $n$ and $k$.
> - The observation that *greedy often tracks influence* is an empirical contribution of our paper – it provides a **first scalable ground truth** benchmark that enables rigorous evaluation of such approximations.
>
> ## Scope and framing
>
> We will revise the introduction and discussion to better align claims with scope.
>
> The paper **does not attempt to solve** combinatorial search or the general MIS problem. It shows that in the linear model, an extremely popular primitive in ML, the combinatorial problem admits an **exact and efficient solution**. Building on this result, we establish consistent set selection for the widely used double/debiased ML under standard assumptions.
>
> We will shorten and moderate the discussion of broader implications (fairness, auditing) to clearly distinguish **what is shown and what is enabled** by the paper.
>
> ### Model class
>
> Our **analysis is limited to partial linear models**, but we disagree with the characterization of this model class as a *narrow subclass* that is *disconnected from modern ML*, or that our discussion overstates *relevance to interpretable ML*.
>
> PLMs are widely used in causal inference and applied statistics, and the DML framework is specifically *designed for modern ML methods* for first-stage nuisance estimation. Least-squares estimation is a foundational primitive in modern ML, and linearized models remain the standard tool in modern *interpretable ML*.
>
> We do not claim applicability to deep non-convex models, and will state this explicitly. We view the result as an algorithmic primitive that is both practically relevant and theoretically tractable.
>
> ### Exactness and estimators
>
> We will clarify the wording on "exactness in RCTs".
> The result applies to the univariate target settings, including canonical RCT estimands computed via difference-in-means or, equivalently, residualized LS.
> These are standard and widely-used estimators in randomized experiments. We do not claim exactness for arbitrary estimators.
>
> ### Interpretability and value of exact set recovery
>
> We are sympathetic to the criticism that interpretability is asserted, and will moderate claims and improve justification.
>
> - The contribution is **methodological**, enabling exact MIS identification where it was previously infeasible. The value of MIS for interpretation is established in the literature we cite.
> - Exact recovery is valuable because it provides the **ground truth optimum**. Without it, the accuracy of approximate methods cannot be assessed beyond toy settings.
> - Our experiments show that non-trivial gaps between greedy approximations and the exact solution can arise. The observation that greedy methods often perform well is itself a substantive empirical finding enabled by our method.
>
> ### Small-sample performance
>
> Our simulations (Section 4.1 and Appendix A5) directly assess finite-sample performance and show that influence estimates are accurate at moderate sample sizes.
> In the univariate setting, the algorithm is *exact in finite samples*; no asymptotic arguments are needed.
>
> ## Originality
>
> The contribution combines existing ideas, which we view as a strength. The novelty lies in integrating ingredients to produce a useful and previously unavailable result – exact $k$-MIS identification in a widely used ML primitive, together with consistency in the well established PML setting.
> This yields an elegant solution to a long-standing computational problem in a way that integrates naturally with existing optimization and semiparametric theory.
> As noted by Konrad & Kuschnig (2026): "Advances in **finding influential sets** remain an active research area, and could substantially improve block-maxima, reduce runtime, and broaden applicability."

---

> > ### Author Rebuttal · Reviewer_Ncj2 · 2026-04-01
> >
> > Can the authors provide figures for the updated experiments as url links? From the existing results, I don't see when greedy can **significantly** deviate from the ground truth in terms of the total score (rather than the set overlap). I think it is the PML setting that makes this difference small. This then leads back to the question of why analyzing PML in the influential set setting is meaningful if the score difference between greedy and optimum is small.

---

> > > ### Author Response · Authors · 2026-04-02
> > >
> > > We thank the reviewer for the follow-up and for raising this important point.
> > >
> > > We agree with the interpretation of the empirical result: in several of our applications, the greedy path tracks the **objective value** of the exact MIS fairly closely for substantial parts of the trace. This *may indeed be a feature of the PLM setting*.
> > > We will make this clearer in the revision and emphasize the following points.
> > >
> > > - First, the **ground-truth** itself is one of the *paper's contributions*. Benchmarking the performance of greedy solutions is only possible because our method makes exact MIS computation feasible at practically relevant sample sizes. Previously, exact results were limited to toy problems, since enumerating MIS for even moderate $k$ is quickly infeasible.
> > >
> > > - Second, the proposed method is *not only exact*, but also **substantially faster** than the greedy approach (e.g., <5 ms versus about 200 ms for $n = 10^4$ and $k = 50$). In regimes such as the microcredit applications, there is no longer a reason to use a greedy approximation in the first place.
> > >
> > > - Third, the paper already features **two counterexamples** showing that the *greedy objective value* **can deviate meaningfully**. (There is no randomness here, so the relevant issue is not statistical significance.)
> > >     - In Figure 1, masking impacts the greedy $3$-MIS and the objective value. The exact 3-MIS moves $\hat\beta = 0.84$ to $0,$ whereas the greedy approximation only reaches $0.3$; more extreme examples are possible.
> > >     - In the Mongolian microcredit trial (Section 4.2.1), *a single non-nestedness event* first changes the set composition substantially (Figure 3, bottom panel), which leads to a gradual divergence in objective value (top panel). The large drop in set overlap ($J < 0.5$) implies that even forward-looking greedy variants would fail.
> > >
> > > We therefore view these results as nuanced.
> > > On the one hand, greedy approximations work reasonably well in the PLM setting.
> > > On the other hand, failures occur, and exact recovery is important, e.g., for validating approximations, revealing failure modes, and worst-case robustness checks, where incorrect sets or understated influence can be misleading.
> > > Our paper addresses these issues for the PLM, while also making clear that further work is needed in more general settings. We will sharpen this discussion in the revision.
> > >
> > > ## AMIP Comparison
> > >
> > > Below, we provide an overview of the promised comparison with AMIP.
> > > A draft figure is available at: <https://upload.disroot.org/r/7_c3VzQf#kneXW2AnQxxR3+wybSi0NMHYNL0f1oSxN3I8w93/HGM=>.
> > >
> > > ### Runtime
> > > Across a range of $(n, k)$ settings, our method is typically *faster to compute than AMIP*, often by an order of magnitude. **Median relative runtime speedups** over 100 runs range from roughly $4 \times$ to more than $50 \times$ in many configurations. The AMIP implementation becomes infeasible for the largest setting we consider in the paper ($n = 10^6$).
> > >
> > > | $n \backslash k$ |    1 |    5 |   10 |   50 |  100 |  500 | 10$^3$ | 10$^4$ | 10$^5$ |
> > > | ---------------: | ---: | ---: | ---: | ---: | ---: | ---: | -----: | -----: | -----: |
> > > | 10 | 21.9 | 22.2 | - | - | - | - | - | - | - |
> > > | 100 | 22.4 | 22.8 | 21.4 | 19.5 | - | - | - | - | - |
> > > | 1000 | 29.7 | 31.8 | 29.5 | 30.8 | 27.5 |  8.8 | - | - | - |
> > > | 10000 | 15.5 | 14.3 | 14.5 | 17.5 | 16.0 |  7.3 |    3.9 | - | - |
> > > | 100000 | 29.9 | 29.5 | 29.7 | 26.2 | 29.0 | 19.0 |   14.5 |    2.8 | - |
> > > | 500000 | 39.1 | 23.9 | 55.0 | 23.7 | 23.5 | 33.8 |   18.1 |    8.4 |    1.1 |
> > >
> > > ### Accuracy
> > > In terms of accuracy, AMIP performs well for small $k$, but its approximation quality deteriorates substantially as $k$ grows.
> > > For example, in the Philippines trial, the **ratio of AMIP influence to exact MIS influence** falls to 0.195 at $k = 500$ and to $0.054$ at $k = 1000$, indicating that AMIP captures only a small fraction of the true maximal influence.
> > >
> > > | Country | Dir. | k=2   | k=10  | k=100 | k=200 | k=500 | k=1000 |
> > > |:------|:---|----:|----:|----:|----:|----:|-----:|
> > > | MON | + | 0.994 | 0.972 | 0.874 | 0.834 | 0.468 |     -  |
> > > | MON | − | 0.994 | 0.989 | 0.914 | 0.614 | 0.250 |     -  |
> > > | PHI | + | 1.000 | 0.998 | 0.755 | 0.460 | 0.195 |  0.054 |
> > > | PHI | − | 0.996 | 0.967 | 0.724 | 0.668 | 0.327 |  0.032 |
> > >
> > > The corresponding **absolute errors, scaled relative to the original coefficient**, reinforce this pattern. In several cases, the discrepancy quickly becomes several times larger than the coefficient itself even for moderate $k$.
> > > This is consistent with existing theoretical results showing that influence-function approximations deteriorate for *sets of observations* and for *highly influential observations*.
> > >
> > > |Country |Dir. | k=2| k=10| k=100| k=200| k=500| k=1000|
> > > |:---|:---|---:|---:|---:|---:|---:|---:|
> > > |MON | + | 0.002| 0.030| 0.40 | 0.64 |  5.1  | -|
> > > |MON | - | 0.001| 0.008| 0.31 | 2.54 | 13.0  | -|
> > > |PHI | + | 0.000| 0.002| 1.20 | 6.02 | 23.9  | 52.3 |
> > > |PHI | - | 0.003| 0.051| 1.11 | 1.82 | 11.1  | 120.3 |

---

### Official Review · Reviewer_BHaU · 2026-03-15

**Soundness:** 3
**Presentation:** 3
**Significance:** 3
**Originality:** 3
**Overall Recommendation:** 4
**Confidence:** 4

**Summary:**

This paper studies how to identify the $k$ data points whose removal changes an estimator the most. Naively, this requires a combinatorial search over all size-$k$ subsets. The paper proposes an algorithm that avoids exhaustive enumeration by exploiting a linear-fractional reformulation of the leave-set-out objective, and it provides theoretical guarantees: exact recovery in the oracle setting and consistency under Neyman orthogonality and first-stage stability in partial linear models.

**Compliance With Llm Reviewing Policy:**

Affirmed.

**Key Questions For Authors:**

- Please answer the questions raised in Strengths And Weaknesses.
- Please clarify the exact asymptotic regime of Theorem 2. Is $k$ fixed, or may it grow with $n$? How do you define the convergence between sets? In particular, if the oracle maximizer is not unique, we need to consider the convergence of sets in sets. In such cases, what exactly is the convergence notion?
- In Proposition 1, is the displayed right-hand side meant to be an influence function in the classical infinitesimal sense, or is it better understood as an exact finite-sample deletion identity? The latter is how it currently reads to me.
- Why are $y$ and $x$ written in lowercase while $Z$ is uppercase? Is $Z$ intended to be vector-valued throughout? If so, please define its dimension and whether $d$ is fixed.
- Please discuss the relationship to orthogonal influence-function work such as Ichimura and Newey, ``The Influence Function of Semiparametric Estimators,'' Quantitative Economics, 2022, and to constructive influence-function work for optimization-based causal functionals such as Jordan, Wang, and Zhou, ``Data-Driven Influence Functions for Optimization-Based Causal Inference,'' arXiv:2208.13701, 2022. In particular, it would be useful to explain more explicitly where the present paper sits relative to (i) classical influence functions, (ii) orthogonal-score corrections, and (iii) data-driven Gateaux-derivative approaches.

**Limitations:**

My main limitation concern is not the relevance of the problem, which I find strong, but the current level of imprecision in the theoretical exposition. At present, the paper reads more convincing as an algorithmic idea than as a fully polished theoretical treatment. I would be much more positive if the authors clarified the mode of convergence in Theorem 2, the role of fixed versus growing $k$, the exact relationship between Proposition 1 and classical influence-function language, and the definitions of joint influence, masking, and greedy path dependence.

**Strengths And Weaknesses:**

The problem is interesting, practical, and potentially important. It is also closely related to the classical literature on robustness and influence diagnostics, including influence curves, deletion diagnostics, masking, and influential subsets, while also being connected to modern semiparametric and machine-learning uses of influence functions.

My understanding is that the present paper approaches the problem from a different angle than classical influence-function methods: instead of relying on infinitesimal or first-order approximations, it seeks exact or consistent recovery of most influential sets by exploiting special structure in the leave-set-out objective. That is potentially a valuable contribution.

That said, the paper currently feels somewhat rough, especially in the precision of the theoretical statements and in the exposition of the motivating concepts.

- Theorem 2 is not stated precisely enough. The limit object is an oracle maximizing set, but the maximizer need not be unique. If the target is a set-valued argmax correspondence, the paper should explicitly define the relevant mode of convergence for random sets or correspondences. If the target is instead a single maximizing set, the paper should state the uniqueness condition up front and keep the notation consistent throughout. Relatedly, the proof sketch seems to operate with $k$ fixed. Is the asymptotic theory meant only for fixed $k$, or can $k = k_n$ grow with $n$? The case $k=n$ seems outside the intended scope, since the leave-set-out estimator degenerates there.
- Section 2.3 motivates the limitations of first-order approximations using terms such as ``joint influence'' and ``masking,'' but these notions are not defined clearly enough. Since they are central to the paper's motivation, I would strongly encourage the authors to define them formally, or at least explain them with a more transparent example and connect them to the classical diagnostics literature.
- Similarly, the greedy-search baseline was not sufficiently clear to me. What exactly is the greedy algorithm? What objective does it optimize at each step? In what precise sense is the objective non-nested or path-dependent? A short pseudo-code description and a more formal discussion of the failure mode would help a lot.
- If the asymptotic argument ultimately relies on an orthogonal asymptotic-linear representation of the second-stage estimator, the paper should explain more sharply what is fundamentally gained beyond first-order methods. My current interpretation is that the gain is not “a better influence function,” but exact finite-sample optimization over subsets when a ratio representation is available, plus consistency under residualization. If that is the right interpretation, I think the paper should say so clearly. This would also make the relation to the debiased-ML literature easier to understand.
- I also found the assumptions behind the asymptotic approximation insufficiently transparent. It is not clear to me which moment, tail, smoothness, or complexity conditions on $y$, $x$, and $Z$ are required for the asymptotic linear representation and the uniform approximation over all $|S|=k$. The notation suggests that $Z$ is vector-valued, but the dimension and asymptotic regime are not made fully explicit. These conditions may well be standard, but they should be stated much more clearly in the main text.
- Proposition 1 seems to be the algebraic core of the paper, yet its proof is deferred entirely to another paper. Since the exact ratio form is foundational to the submission, a short self-contained derivation would materially improve readability and confidence in the argument.

---

> ### Author Rebuttal · Authors · 2026-03-31
>
> We thank the reviewer for their careful reading and constructive feedback. We will revise the paper to sharpen the precision of results, clarify motivating concepts, and make assumptions more transparent.
>
> ## Contribution & positioning
>
> The reviewer's interpretation is correct.
> The goal is not to refine infinitesimal influence analysis, but to identify the size-$k$ subset whose removal changes the estimator the most.
> The key is that, for the estimators we study, the leave-set-out objective admits an *exact linear-fractional representation*. This enables us to solve the finite-sample combinatorial optimization problem *exactly* in the oracle setting, and to establish consistent set selection for partial linear models.
> The gain over classical approaches is **exact combinatorial optimization** when a ratio representation is available, along with consistency under residualization.
> We will make this positioning more explicit.
>
> ## Theorem 2: maximizers & asymptotics
>
> We agree that Theorem 2 is currently too imprecise and will revise it accordingly.
>
> #### Uniqueness
> The main result assumes **a unique oracle maximizer** $\mathbb{S}_k^{\mathrm{or}}$, yielding selection consistency:
> $$
>     \Pr \left( \hat{ \mathbb{S} }_k^{\max} = \mathbb{S}_k^{\mathrm{oracle}} \right) \to 1.
> $$
> This follows from uniform convergence of the objective and a gap condition.
>
> If maximizers are **non-unique**, let $\mathbb{M}_k^{\mathrm{oracle}}$ be the argmax set. The appropriate convergence statement is:
> $$
>   \Pr \left(\hat{\mathbb{S}}_k^{\max} \in \mathbb{M}_k^{\mathrm{oracle}}\right) \to 1.
> $$
> We will state both cases explicitly, while keeping uniqueness as the main result, since it holds almost surely under continuous distributions and requires no tie-breaking.
>
> #### Asymptotics
> The theorem is stated for **fixed** $k$; we will make this explicit.
>
> We are also developing an extension to *growing* $k$. Our analysis suggests that $k_n = o(n^{1/2})$ is sufficient without stronger assumptions on nuisance convergence; the binding constraint arises from the numerator approximation.
> We will include this as corollary if finalized.
>
> The $k = n$ case is excluded, because the objective degenerates; this is ruled out by the positivity condition $G(\mathbb{S}) > 0.$
>
> #### Assumptions
> We will move the **key regularity conditions** into the main text: i.i.d. sampling, conditional exogeneity, bounded conditional variance, non-degenerate treatment variation, and sufficient rates/stability of nuisance learners.
> These are standard in the DML literature; we will state them explicitly and explain their roles.
>
> We will also clarify notation: lowercase $y_i, x_i$ are scalar outcome and treatment, while $Z_i$ denotes the **control vector**.
> The result is stated for $Z_i \in \mathbb{R}^d$ *with fixed dimension* $d,$ so the smoothness condition $s > d/2$ on $m_0$ and $h_0$ is sufficient for the product-rate condition $\alpha_m + \alpha_h > 1/2.$
> Growing $d$ is compatible with our results as long as the nuisance estimators achieve the required rate (e.g., under sparsity).
>
> ### Proposition 1
>
> The reviewer's reading is correct – Proposition 1 is an exact finite-sample deletion identity.
> We will clarify this and include a short, self-contained derivation (see the response to Reviewer EEDE#4).
>
> ### Joint influence, masking, & greedy search
>
> We agree that these concepts deserve clearer definitions and exposition and will:
>
> - Define *joint influence* as $ \Delta( \mathbb{S} ) \gg \sum_{s \in \mathbb{S}} \Delta(s),$ arising from the score–curvature trade-off.
> - Define *masking* through non-nested maximizers, e.g., $\mathbb{S}\_{k-1}^{\max} \not \subset \mathbb{S}\_{k}^{\max},$ so local approximations can miss the $k$-MIS.
> - State the *greedy algorithm* explicitly: (0) Initialize at $\mathbb{S}\_{0} = \varnothing,$ then for $j = 1, \dots, k$ form $\mathbb{S}\_{j} = \mathbb{S}\_{j-1} \cup \{i\},$ where $i \not\in \mathbb{S}\_{j-1}$ is the most influential observation *after* removing $\mathbb{S}\_{j-1}$. We will add pseudocode to the Appendix.
> - Expand the example to illustrate how joint influence and path dependence cause greedy approximations to fail.
>
> ### Relationship to influence functions and DML
>
> Thank you for these references; we will incorporate them in the revision. We see the relation as follows:
>
> - Classical and semiparametric *influence functions* (e.g., *Ichimura-Newey*) characterize infinitesimal perturbations.
> - Orthogonal/DML methods use these to obtain *first-order robustness* with respect to nuisance estimators.
> - Constructive approaches (e.g., *Jordan-Wang-Zhou*) approximate local derivative information when analytical derivations are unavailable.
>
> By contrast, our method uses an *exact finite-sample identity* to solve a *discrete subset selection problem*. Orthogonality enters only to justify residualization; the main contribution is handling finite, non-local deletions via exact optimization (oracle) and consistent recovery after residualization.

---

> > ### Author Rebuttal · Reviewer_BHaU · 2026-04-08
> >
> > Thank you for your reply. I now have a better understanding of most of my questions. As for the technical issues, I haven’t had time to fully digest the authors’ responses. I will proceed with verification within the review period.

---

### Decision · Program_Chairs · 2026-04-30

**Decision:**

Accept (regular)

**Comment:**

This paper proposes a new algorithm for finding *exact* most influential size-$k$ sets, for a class of (univariate) estimands such that the MIS problem can be phrased in terms of a linear-fractional program. Under this perspective, the problem can be solved using Dinkelbach's algorithm instead of the naive brute-force search.

Reviewers generally appreciated the problem reformulation, and the guarantee that the algorithm returns an **exact** MIS instead of a heuristic or approximate solution. However, there is also concern regarding the comparison with the greedy heuristic baseline. In the current presentation, the experiments do not demonstrate a sufficient separation of greedy vs the exact MIS results (e.g. even in the Mongolian dataset, there is separation only in limited regimes), which appears to detract from the theoretical claims of the significance of the exactness guarantee.

Given that the authors are allowed to pick which benchmarks/datasets to present, it would significantly strengthen the paper narrative by giving more examples of stronger separation between the methods in the objective value of the estimand (and not just the set composition).